# Mitigating Embedding Leakage via Latent Disruption with Controlled Reconstruction

**Zhiyuan Wu**                                                                          *zhiyuanw@uio.no*
*University of Oslo*

**Changkyu Choi**                                                                       *changkyc@ifi.uio.no*
*University of Oslo*

**Shujian Yu**                                                                          *s.yu3@vu.nl*
*Vrije Universiteit Amsterdam*
*UiT – The Arctic University of Norway*

**Robert Jenssen**                                                                      *robert.jenssen@uit.no*
*UiT – The Arctic University of Norway*
*University of Copenhagen*
*Norwegian Computing Center*

**Ali Ramezani-Kebrya**                                                                 *ali@ifi.uio.no*
*University of Oslo*
*Integreat – Norwegian Centre for Knowledge-driven Machine Learning*
*TRUST – The Norwegian Centre for Trustworthy AI*

**Reviewed on OpenReview:** *https://openreview.net/forum?id=nZWBrxJyrS*

## Abstract

Pre-trained encoders produce semantically rich latent embeddings, which, however, may expose unintended information through malicious inference or exploitation. We propose **SEAL**, a framework that mitigates embedding leakage by disrupting latent representations based on information-theoretic principles. It reduces the risk of potential misuse while enabling *controlled* reconstruction for trusted users. **SEAL** learns to encode controlled perturbations by minimizing the *Matrix Norm-based Quadratic Mutual Information* (MQMI) functional between original and perturbed embeddings within a hyperspherical latent space. Meanwhile, a private decoder, jointly trained with the **SEAL** encoder, is trained to reconstruct the original data that is accessible only to authorized users under an access-controlled setting. Extensive experiments on vision and text datasets demonstrate that **SEAL** reduces latent leakage, weakens the effectiveness of evaluated inference attacks, and preserves reconstruction under the considered setting.

## 1 Introduction

Large-scale pretrained encoders have transformed data-sharing workflows through latent representations that are semantically rich, transmission-efficient, yet not directly human-interpretable (Awais et al., 2025; Zhang et al., 2024; Huang et al., 2024; Oquab et al., 2024; Nobi et al., 2022; Dosovitskiy et al., 2021). Meanwhile, in scenarios that involve human-in-the-loop interpretation under access-controlled settings, such as clinical diagnosis based on medical data (Olawade et al., 2026), human oversight relies on access to semantically meaningful information to support interpretation. Retaining such semantic information in shared representations, however, introduces a potential risk of misuse, as latent embeddings may be exposed beyond the controlled boundary. This motivates the study of how to empirically mitigate misuse of such embeddings

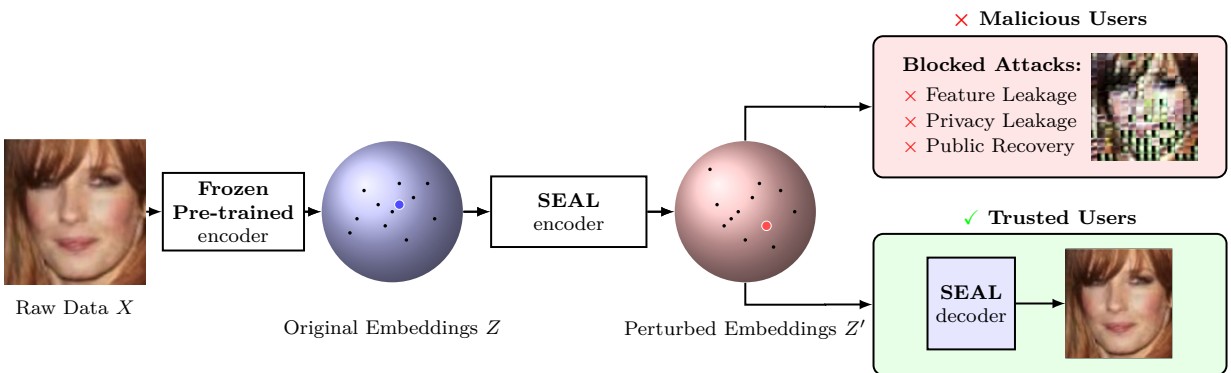

Figure 1: Overview of the proposed **SEAL** framework. A pretrained encoder, such as Transformer-based models (Siméoni et al., 2025; Oquab et al., 2024; Caron et al., 2021b; Warner et al., 2024; Liu et al., 2019; Devlin et al., 2019), maps sensitive data $X$ to latent embeddings $Z$ on a hyperspherical manifold (illustrated here as a 2D sphere for clarity), where the blue dot represents a sample among other data embeddings (black dots). A *controlled transformation*, learned by the **SEAL** encoder, perturbs $Z$ into the embedding $Z'$, shown as the red dot. Reconstruction through the **SEAL** decoder is available to trusted users with access.

under a threat scenario where they may be accessible to untrusted parties, while mechanisms enabling human interpretation remain available only to authorized users.

Latent embeddings have been increasingly recognized as potential sources of privacy leakage (Huang et al., 2024; Morris et al., 2023; Samuel et al., 2023; Kwon et al., 2023; Fredrikson et al., 2015). Leaked embeddings can be maliciously exploited to retrieve sensitive information (He et al., 2025), or attacked through membership and attribute inference (Wen et al., 2024; Kim et al., 2024; Ko et al., 2023; Jia & Gong, 2018).

To mitigate potential misuse under data leakage in the input space, *data poisoning* methods introduce imperceptible perturbations to hinder unauthorized model training (Fowl et al., 2024; Fang et al., 2024; Zhu et al., 2024; Liu et al., 2024; Meng et al., 2024; He et al., 2023; Liu et al., 2023; Huang et al., 2021; Biggio & Roli, 2018). However, these approaches mitigate risks at the input level and are not designed to address the risks posed by latent embeddings once they are leaked, under our threat scenario setting.

Recent work has shown that slight conditional perturbations of embeddings during pre-training can improve the generative performance of diffusion models, where such perturbations are typically studied under benign learning objectives as a form of regularization (Chen et al., 2024; Ho et al., 2020). In contrast, under our threat model, we consider an adversary who has access to leaked latent embeddings, but does not have access to the private reconstruction decoder. The adversary aims to extract exploitable information from these embeddings through downstream inference and reconstruction tasks, including learning predictive mappings from embeddings, distinguishing training samples from unseen ones, recovering sensitive attributes (Hu et al., 2022; Shokri et al., 2017; Jia & Gong, 2018), and reconstructing the original data using a public decoder. Our goal is to empirically mitigate such risks through controlled perturbations while preserving sufficient information for authorized, human-in-the-loop interpretation.

We propose **SEAL**, named to reflect the goal of mitigating embedding leakage. It learns to introduce controlled perturbations in the latent space and co-trains a decoder that enables authorized reconstruction from perturbed embeddings back to the data domain. The perturbation mechanism is guided by a tractable surrogate objective for controlling dependency between representations, while jointly optimizing a reconstruction objective through the decoder. As a result, the shared embeddings are less suitable for downstream exploitation, while meaningful information remains accessible only through decoder-based reconstruction, with the decoder acting as part of the access-controlled mechanism. **SEAL** consists of a encoder and a decoder, as illustrated in Figure 1.

The main contributions of our work are:

- We propose **SEAL**, a framework that learns to introduce controlled perturbations to latent embeddings, *empirically reducing the risk of misuse under embedding leakage* while enabling *controlled reconstruction* for trusted users through a jointly trained **SEAL** decoder.

- **SEAL** introduces the *Matrix Norm-based Quadratic Mutual Information (MQMI)*, a *tractable surrogate objective* defined over the eigenspectrum of kernel matrices for controllable dependency regulation between original and perturbed embeddings, enabling interpretable control over the disruption process.

- Experiments show that **SEAL** reduces the effectiveness of evaluated misuse and inference attacks of embeddings, such as training downstream classifiers or $k$-NN models, while reducing risks of membership and attribute inference. It simultaneously preserves reconstruction quality for trusted users through the **SEAL** decoder.

## 2 Related Work

### 2.1 Data Protection through Controlled Corruption

Data poisoning attacks (Zhu et al., 2024; Liu et al., 2024; Meng et al., 2024; He et al., 2023; Zhang et al., 2023), also referred to as data availability poisons (Fang et al., 2024; Liu et al., 2023), aim to protect data from malicious use by deliberately corrupting datasets, thereby preventing models trained on them from generalizing effectively. For support vector machines (SVMs), even a single poisoned data point could degrade model performance significantly (Biggio & Roli, 2018; Demontis et al., 2019; Biggio et al., 2012). These attacks are generated through bi-level optimization (Mei & Zhu, 2015; Xiao et al., 2015; Biggio et al., 2012).

In deep learning, the aforementioned optimization becomes computationally infeasible, motivating data poisoning attacks like *unlearnable examples* (Zhu et al., 2024; Liu et al., 2024; Meng et al., 2024; Zhang et al., 2023; Huang et al., 2021) and adversarial poisoning (Fowl et al., 2024) that makes strong poisons. These methods typically use Projected Gradient Descent (PGD) (Fowl et al., 2024; Huang et al., 2021; Madry et al., 2017) to inject perturbations in the input space. By adding carefully crafted perturbations imperceptible to humans, these methods produce datasets that are unexploitable for model training.

Recent research has also extended data poisoning attacks to self-supervised frameworks like SimCLR (Chen et al., 2020), MoCo (He et al., 2019), and BYOL (Grill et al., 2020), targeting contrastive learning objectives. Despite the strong representation learning capability of contrastive frameworks, data poisoning can still produce datasets unexploitable for contrastive training and significantly degrade performance on downstream tasks (He et al., 2023).

### 2.2 The Risk of Malicious Exploitation in Latent Spaces

Data poisoning methods mainly perturb source data, such as images (Liu et al., 2024; Fang et al., 2024; Huang et al., 2021). However, latent representations extracted by publicly available encoders, such as a DINO family (Siméoni et al., 2025; Oquab et al., 2024; Caron et al., 2021b) and a BERT family (Warner et al., 2024; Liu et al., 2019; Devlin et al., 2019) have emerged as a critical source of privacy leakage (Huang et al., 2024; Morris et al., 2023; Samuel et al., 2023; Kwon et al., 2023; Fredrikson et al., 2015).

In practical deployments such as healthcare or enterprise collaboration (Schneider et al., 2024; Khalid et al., 2023), data-driven workflows increasingly rely on information across systems and services (Bhatta, 2024; Borra, 2024; Zou et al., 2020). As such information propagates across services, latent embeddings that encode high-level semantics become particularly vulnerable to unintended exposure.

Recent studies show that embeddings from pretrained models retain substantial semantic content, making them exploitable even without access to original inputs (Huang et al., 2024; Morris et al., 2023; Fredrikson et al., 2015). Feature inversion attacks (Fredrikson et al., 2015; Mahendran & Vedaldi, 2014) pose a major threat by reconstructing approximate inputs or inferring sensitive attributes directly from latent spaces. These threats motivate the need for protection mechanisms against malicious exploitation of latent embeddings.

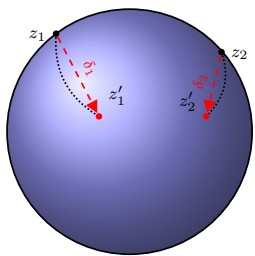

Figure 2: Visualization of hyperspherical disruption. Two example points $z_1$ and $z_2$ and their perturbed versions $z_1'$, $z_2'$ (red dots) are shown on a 2D sphere. Red dashed lines denote chord distances $\delta_1$ and $\delta_2$, while black dotted curves indicate geodesic paths on the hypersphere.

## 3 Method

The geometry of latent embeddings is determined by the input data distribution and the encoder architecture. In typical practical settings, e.g., transformers with LayerNorm (Wu et al., 2024; Ba et al., 2016), embeddings exhibit high-probability concentration near a hypersphere $\|z\|_2 \simeq R$ for some radius $R$. This behavior is consistent with concentration of norms in high dimensions (Vershynin, 2018)(see, e.g., Chapter 3, in particular Section 3.1 on concentration of the norm), and is also observed empirically across pretrained models and datasets, as illustrated in Figure 8 in Appendix F.

### 3.1 Setup and Objective

We assume $\|z\|_2 \simeq R$, and train a **SEAL** encoder $h_\theta : \mathbb{R}^d \to \mathbb{R}^d$ to map each embedding $z$ to its perturbed counterpart $z' = h_\theta(z)$, while preserving the hyperspherical constraint. We define the *chordal shift* $\delta = z' - z$, as illustrated in Figure 2, and denote by $Z$ and $Z'$ the corresponding random variables. We aim to control the statistical dependence between $Z$ and $Z'$, such that the **SEAL** encoder learns to introduce controlled disruption over the hyperspherical embedding manifold.

### 3.2 Matrix Norm-based Quadratic Rényi's Entropy Functional

Estimating mutual information is notoriously difficult, particularly in high-dimensional settings (Czyz et al., 2025; 2023; Goldfeld & Greenewald, 2021). While variational approaches based on neural networks require auxiliary model training, we adopt recent spectral methods (Skean et al., 2024; Yu et al., 2019; Giraldo et al., 2014) that define entropy functional directly over the eigenspectrum of kernel matrices. This avoids explicit probabilistic density modeling in high-dimensional spaces, satisfies Rényi's axioms for entropy (Rényi, 1961), and allows for efficient, training-free computation. Given a batch of $N$ embedding samples $\{z_i\}$ from the hyperspherical random variable $Z$, we construct the Gram matrix $K^Z \in \mathbb{R}^{N \times N}$ to capture pairwise similarities.

For hyperspherical embeddings, the von Mises–Fisher (vMF) kernel is commonly used to model directional data (Trosten et al., 2023; Wang & Isola, 2022). Specifically, the vMF kernel entry between two embeddings $z_i$ and $z_j$ is given by $K^Z_{ij} = \exp\left(\kappa\, z_i^\top z_j / (\|z_i\|_2 \|z_j\|_2)\right)$, where $\kappa > 0$ is a concentration hyperparameter controlling the sensitivity of the similarity measure.

In the following, we define a normalized positive definite (NPD) matrix $A^Z$, obtained from the Gram matrix $K^Z$ evaluated with the vMF kernel, as

$$A^Z = \frac{K^Z}{\mathrm{Tr}(K^Z)}, \tag{1}$$

following the convention in Yu et al. (2019); Giraldo et al. (2014).

### 3.2.1 Matrix Norm-based Quadratic Entropy Functional

The *Matrix Norm-based Quadratic (MQ)* entropy functional of $Z$ is given by:

$$\mathbf{S}_2(Z) \;=\; -\log \big\|A^Z\big\|_F^2, \tag{2}$$

where $A^Z = K^Z/\mathrm{Tr}(K^Z)$ is the normalized Gram matrix. The Frobenius norm satisfies $\|A^Z\|_F^2 = \mathrm{Tr}\big((A^Z)^2\big) = \sum_i \lambda_i^2$, where $\{\lambda_i\}$ are the eigenvalues of $A^Z$, connecting the formulation to the spectral definition of matrix-based entropy. The formulation follows the eigenvalue-based definition of the matrix-based Rényi's entropy functional. By restricting to the second-order (quadratic) form, the expression naturally avoids the eigen decomposition of the normalized Gram matrix, thereby reducing the computational complexity from $O(N^3)$ to $O(N^2)$ (Skean et al., 2025; 2024; Bach, 2022; Yu et al., 2019).

We define the perturbation variable $\Delta := Z' - Z$, where each chord $\delta_i = z_i' - z_i$ connects corresponding points $z_i$ and $z_i'$ on the hypersphere. Since the perturbations $\delta_i$ are defined as Euclidean differences, rather than geodesic distances on the hypersphere, it is natural to measure their similarity using a kernel defined over Euclidean distances. We therefore model the structure of $\Delta$ using a radial basis function (RBF) kernel.

We compute the Gram matrix $K^\Delta$ over samples $(\delta_1, \delta_2, \ldots, \delta_N)$. Specifically, the entries are given by $K_{ij}^\Delta = \exp\big(-\|\delta_i - \delta_j\|_2^2/(2\sigma^2)\big)$, where $\sigma$ is the kernel width parameter. The MQ entropy functional of $\Delta$ is then defined as:

$$\mathbf{S}_2(\Delta) \;=\; -\log \big\|A^\Delta\big\|_F^2, \tag{3}$$

which evaluates the MQ entropy functional on the chordal perturbations in the latent space.

### 3.2.2 MQ Joint Entropy Functional

We define the *joint Gram matrix* $K^{Z,Z'}$ as the element-wise (Hadamard) product of the individual Gram matrices (Yu et al., 2019), that is, $K^{Z,Z'} = K^Z \odot K^{Z'}$. The MQ joint entropy functional of $Z, Z'$ is given by:

$$\mathbf{S}_2(Z, Z') \;=\; -\log \big\|A^{Z,Z'}\big\|_F^2. \tag{4}$$

### 3.3 Matrix Norm-based Quadratic Mutual Information Functional

We aim to quantify the dependency between the original embeddings $Z$ and their perturbed counterparts $Z'$. In analogy with Shannon's definition, we adopt a matrix-based functional of *Quadratic Mutual Information* (Yu et al., 2019):

$$\mathbf{I}_2(Z, Z') = \mathbf{S}_2(Z) + \mathbf{S}_2(Z') - \mathbf{S}_2(Z, Z'). \tag{5}$$

This defines a matrix-based analogue of quadratic mutual information, formulated over kernel Gram matrices.

While the exact form of $\mathbf{S}_2(Z')$ depends jointly on both the original embeddings and the perturbations, it is desirable to separate the effect of the perturbation in order to explicitly regulate it.

To enable direct control over how perturbations affect the representations, we refer to the chordal shift variable $\Delta$, which captures how each embedding moves on the hypersphere. Instead of modeling the marginal entropy of $Z'$ directly, we regulate the perturbation statistics through the MQ entropy of $\Delta$, allowing **SEAL** to shape the perturbation and its interaction with the original embeddings.

Based on this reformulation, we construct a surrogate objective inspired by the matrix-based mutual information functional, where the effect of the perturbation is explicitly controlled through $\Delta$, rather than directly modeling $\mathbf{S}_2(Z')$.

This yields the following objective:

$$\mathrm{MQMI}(Z, Z') = -\beta \underbrace{\log \|A^\Delta\|_F^2}_{\text{entropy}} + \underbrace{\log \|A^{Z,Z'}\|_F^2}_{\text{disruption}}, \tag{MQMI}$$

where $\beta \in \mathbb{R}$ acts as a Lagrange multiplier balancing:

- **Entropy:** the MQ marginal entropy of the perturbations $\Delta$;

- **Disruption:** the MQ joint entropy between $Z$ and $Z'$.

MQMI is a surrogate objective instead of a mutual information estimator with statistical guarantees. It consists of two terms that jointly shape the perturbation. Its justification and the conditions under which replacing $\mathbf{S}_2(Z')$ with $\mathbf{S}_2(\Delta)$ is meaningful are discussed in subsequent sections and Appendix D.

### 3.4 Joint Training with SEAL Decoder

By minimizing MQMI, we empirically disrupt the effectiveness of evaluated malicious exploitation of embeddings.

To enable controlled reconstruction, we train a private decoder jointly with the perturbation mechanism to recover the original data from perturbed embeddings. This ties the decoder to a perturbation-specific inverse mapping, allowing reconstruction for authorized users, while making it difficult to recover faithful data from perturbed embeddings alone.

We jointly optimize the **SEAL** encoder $h_\theta$, which generates perturbed embeddings $Z'$, and the **SEAL** decoder $g_\omega$, which enables controlled reconstruction $\hat{X} = g_\omega(Z')$, as illustrated in Figure 1. The overall objective is:

$$\min_{\theta,\omega} \left[ \mathrm{MQMI}(Z, Z') \; + \; \gamma \, \mathcal{L}_{\mathrm{recon}}(\hat{X}, X) \right], \tag{SEAL}$$

where $\mathcal{L}_{\mathrm{recon}}(\hat{X}, X)$ enforces reconstruction through the **SEAL** decoder using perturbed embeddings. In practice, $\mathcal{L}_{\mathrm{recon}}$ can be instantiated with standard reconstruction losses (e.g., mean squared error), and can be adapted to application-dependent objectives. The hyperparameter $\gamma$ controls the trade-off between disruption and reconstruction.

### 3.5 Geometric Motivation for the Perturbation Term

We provide a geometric motivation for introducing the perturbation term $\mathbf{S}_2(\Delta)$ in our objective.

Since both the original embeddings $Z$ and the perturbed embeddings $Z'$ lie on $\mathbb{S}^{d-1}$, their difference can be represented as a chordal shift

$$\Delta := Z' - Z.$$

For each sample, the spherical constraint $\|z_i'\|_2 = 1$ implies

$$z_i^\top \delta_i = -\tfrac{1}{2} \|\delta_i\|_2^2,$$

which indicates that the perturbation $\delta_i$ is not arbitrary but constrained by the spherical geometry.

This suggests that the perturbation follows an intrinsic geometric structure. A natural question is whether such geometric changes can be reflected at the information level, i.e., whether the effect from $Z$ to $Z'$ can be decomposed into a perturbation-dependent component and a part inherent to $Z$.

The entropy functional $\mathbf{S}_2(\cdot)$ encodes pairwise similarities through kernel evaluations. Applying it to $\Delta$ provides a natural way to characterize perturbation-induced changes at the level of information functionals, rather than measuring them jointly with the original embeddings through the perturbed representations $Z'$. This serves as a geometric motivation; a more detailed analysis, including the underlying assumptions and approximation steps, is provided in Appendix D.

## 4 Experiments

We present a comprehensive evaluation of our proposed **SEAL** framework, demonstrating its ability to disrupt latent representations against malicious exploitation while preserving faithful reconstruction for authorized users. We use STL-10 (Deng et al., 2009), Tiny-ImageNet (Deng et al., 2009), CelebA-Smiles (Liu et al., 2015),

| STL-10 | | | | | | Tiny-ImageNet | | | | |
|---|---|---|---|---|---|---|---|---|---|---|
| Metric | Baseline | MMD | CS-Div | CS-QMI | MQMI | Metric | Baseline | MMD | CS-Div | CS-QMI | MQMI |
| LR ↓ | 95.45 | 86.16 | 78.91 | 36.10 | **24.59** | LR ↓ | 77.78 | 77.74 | 76.82 | 78.78 | **74.70** |
| R@1 ↓ | 84.06 | 80.35 | 80.16 | 15.90 | **1.34** | R@1 ↓ | 74.62 | 74.92 | 72.62 | 64.84 | **31.62** |
| R@5 ↓ | 96.79 | 95.61 | 95.73 | 47.43 | **4.79** | R@5 ↓ | 87.84 | 88.10 | 87.10 | 82.88 | **45.66** |
| F1@5 ↓ | 86.47 | 84.18 | 84.10 | 23.90 | **1.88** | F1@5 ↓ | 78.16 | 79.06 | 77.37 | 70.84 | **32.21** |
| MLP ↓ | 94.75 | 72.54 | 75.04 | 32.42 | **10.39** | MLP ↓ | 78.56 | 76.88 | 76.64 | 77.46 | **53.82** |

| 20 Newsgroup | | | | | | AG News | | | | |
|---|---|---|---|---|---|---|---|---|---|---|
| Metric | Baseline | MMD | CS-Div | CS-QMI | MQMI | Metric | Baseline | MMD | CS-Div | CS-QMI | MQMI |
| LR ↓ | 61.47 | 28.41 | 5.57 | 12.02 | **3.09** | LR ↓ | 90.53 | 4.52 | **2.70** | 68.84 | 5.04 |
| R@1 ↓ | 41.10 | 21.79 | 4.02 | 11.14 | **3.62** | R@1 ↓ | 86.25 | 38.24 | **4.67** | 41.73 | 11.50 |
| R@5 ↓ | 72.12 | 50.42 | 16.70 | 39.02 | **15.38** | R@5 ↓ | 97.38 | 66.53 | **14.38** | 81.54 | 37.70 |
| F1@5 ↓ | 47.04 | 29.31 | 6.96 | 18.90 | **5.46** | F1@5 ↓ | 89.77 | 44.07 | **7.58** | 53.24 | 17.82 |
| MLP ↓ | 64.12 | 28.99 | 16.88 | 34.03 | **2.65** | MLP ↓ | 91.47 | **4.94** | 5.20 | 72.64 | 11.55 |

Table 1: **Feature Leakage Evaluation.** We evaluate by training models on secured embeddings and testing on clean embeddings. The baseline uses clean embeddings for both training and testing. Lower values (↓) indicate stronger disruption. Classification and retrieval performance are measured using logistic regression and $k$-NN across vision (STL-10 with ViT-Base (MAE), Tiny-ImageNet with ViT-Base (DINO)) and text (20 Newsgroup, AG News with BERT-Base) datasets. Metrics include classification accuracy (LR), recall at $k$ (R@1, R@5), F1-score. We compare the proposed MQMI with other kernel-based methods: MMD, CS divergence (CS-Div), and CS-QMI.

| UTKFace | | | | | | CelebA-Smiles | | | | |
|---|---|---|---|---|---|---|---|---|---|---|
| Metric | Base. | MMD | CS-Div | CS-QMI | MQMI | Metric | Base. | MMD | CS-Div | CS-QMI | MQMI |
| M-Acc↓ | 50.15 | 50.05 | **49.93** | 50.15 | 50.96 | M-Acc↓ | 50.02 | 49.82 | 49.94 | **49.62** | 49.87 |
| M-Prec↓ | 80.16 | 80.05 | **79.94** | 80.16 | 80.96 | M-Prec↓ | 80.02 | 79.82 | 79.94 | **79.62** | 79.87 |
| M-AUC↓ | 0.51 | 0.50 | 0.50 | 0.51 | 0.52 | M-AUC↓ | 0.50 | 0.50 | 0.50 | 0.50 | 0.50 |
| $A_G$-Acc↓ | 91.31 | 90.59 | 90.51 | 91.25 | **81.97** | $A_M$-Acc↓ | 96.93 | 96.63 | 96.73 | 96.54 | **92.01** |
| $A_G$-Prec↓ | 91.32 | 90.60 | 90.55 | 91.25 | **80.96** | $A_M$-Prec↓ | 96.80 | 96.49 | 96.60 | 96.38 | **91.79** |
| $A_G$-AUC↓ | 0.97 | 0.97 | 0.97 | 0.97 | **0.90** | $A_M$-AUC↓ | 0.99 | 0.99 | 0.99 | 0.99 | **0.97** |
| $A_E$-Acc↓ | 77.47 | 78.51 | **77.16** | 77.39 | 77.24 | $A_A$-Acc↓ | 80.95 | 80.83 | 80.88 | 80.68 | **79.39** |
| $A_E$-Prec↓ | 68.04 | 70.22 | 69.32 | **67.57** | 68.01 | $A_A$-Prec↓ | 80.98 | 80.85 | 80.91 | 80.69 | **79.39** |
| $A_E$-AUC↓ | 0.91 | 0.92 | 0.91 | 0.91 | 0.90 | $A_A$-AUC↓ | 0.90 | 0.89 | 0.90 | 0.89 | **0.87** |

| 20 Newsgroup | | | | | | AG News | | | | |
|---|---|---|---|---|---|---|---|---|---|---|
| Metric | Base. | MMD | CS-Div | CS-QMI | MQMI | Metric | Base. | MMD | CS-Div | CS-QMI | MQMI |
| M-Acc↓ | 56.79 | 52.78 | **52.33** | 57.53 | 57.59 | M-Acc↓ | 50.06 | 50.30 | 50.46 | 50.04 | **49.99** |
| M-Prec↓ | 66.83 | 62.81 | **62.37** | 67.57 | 67.62 | M-Prec↓ | 94.10 | 94.34 | 94.50 | 94.08 | **94.03** |
| M-AUC↓ | 0.63 | **0.54** | **0.54** | 0.63 | 0.62 | M-AUC↓ | 0.50 | 0.52 | 0.53 | 0.50 | 0.50 |
| A-Acc↓ | 58.86 | 58.76 | 56.70 | 56.72 | **55.56** | A-Acc↓ | 90.32 | 87.89 | **84.67** | 89.47 | 89.26 |
| A-Prec↓ | 58.51 | 57.12 | 55.17 | 55.99 | **54.42** | A-Prec↓ | 90.31 | 87.90 | **84.70** | 89.45 | 89.27 |
| A-AUC↓ | 0.94 | 0.93 | **0.92** | 0.93 | 0.93 | A-AUC↓ | 0.98 | 0.98 | **0.96** | 0.98 | 0.98 |

Table 2: **Privacy Evaluation.** We evaluate membership (M) and attribute (A) inference attacks under various disruption methods. In each attack, a classifier is trained and evaluated on secured embeddings to simulate latent leakage. Metrics include accuracy, precision, and AUC, where lower values (↓) indicate stronger privacy preservation. Vision benchmarks use DINO-Base ViT encoders and test for gender ($A_G$, $A_M$), ethnicity ($A_E$), and attractiveness ($A_A$); text benchmarks use BERT-Base and evaluate for category-level attribute prediction.

and UTKFace (Zhang et al., 2017) for vision tasks, and Emotion (Saravia et al., 2018), AG News (Zhang et al., 2015), and 20 Newsgroups (Mitchell, 1997) for text tasks. Vision data uses ViT-base (MAE) (He

| STL-10 | LR ↓ | R@1 ↓ | R@5 ↓ | F1@5 ↓ | | Emotion | LR ↓ | R@1 ↓ | R@5 ↓ | F1@5 ↓ |
|---|---|---|---|---|---|---|---|---|---|---|
| $\beta = 0.0$ | 45.45 | 41.10 | 71.14 | 46.98 | | $\beta = 1.0$ | 57.85 | 37.35 | 79.55 | 50.04 |
| $\beta = -0.1$ | 34.94 | 40.85 | 71.07 | 46.50 | | $\beta = 0.0$ | 50.25 | 18.70 | **51.35** | 32.87 |
| $\beta = -1.0$ | **24.59** | **1.34** | **4.79** | **1.88** | | $\beta = -1.0$ | 12.80 | 20.00 | 66.95 | 30.45 |
| **CS-QMI** | 36.10 | 15.90 | 47.43 | 23.90 | | **CS-Div** | **10.90** | **15.90** | 60.20 | **27.15** |

Table 3: **Ablation Study on $\beta$ for MQMI.** Left: Results on STL-10 using ViT-Base (MAE). Right: Results on Emotion dataset using BERT-Base. Negative $\beta$ values encourage stronger disruption. The explored range of $\beta$ differs across datasets and is chosen to cover a representative range in each setting.

| $\gamma$ | 0.1 | 0.5 | 1.0 | 2.0 | 5.0 |
|---|---|---|---|---|---|
| LR ↓ | 70.90 | 78.23 | 71.61 | 80.49 | 82.71 |
| R@1 ↑ | 33.55 | 37.92 | 39.55 | 38.49 | 39.85 |
| R@5 ↑ | 72.15 | 71.25 | 71.43 | 71.50 | 71.88 |
| F1@5 ↑ | 44.25 | 46.12 | 45.68 | 45.88 | 46.82 |

Table 4: **Ablation Study on $\gamma$ for MQMI.** LR: leakage rate (%), R@k: retrieval accuracy (%), and F1@5: F1 score over top-5 retrievals. Lower LR and higher R/F1 indicate better performance.

et al., 2021) and ViT-base (DINO) (Caron et al., 2021a) as the encoder, while text data uses a BERT-base model (Devlin et al., 2019).

We use vMF kernels for hyperspherical embeddings and RBF kernels for Euclidean perturbations $\Delta$, reflecting the underlying geometry of the representations. The concentration parameter $\kappa$ for the vMF kernel is selected via grid search, while the bandwidth for the RBF kernel is determined using the median heuristic.

The trade-off parameters $\beta$ (perturbation strength) and $\gamma$ (reconstruction weight) control the balance between reducing exploitable information and preserving reconstruction quality, and are selected via validation-based grid search over a small range. All backbone encoders are kept frozen, and only the **SEAL** encoder and the **SEAL** decoder are trained. Token representations are chosen consistently with the backbone architectures (we primarily use the [CLS] token), with further details provided in Appendix C. Full experimental details are provided in Appendices A and B.

## 4.1 Feature Leakage Analysis

We evaluate feature leakage by assessing how well perturbed latent embeddings can be exploited to generalize to clean embeddings. Specifically, we simulate an attacker who obtains leaked perturbed embeddings, trains models on them, and tests on clean embeddings to measure the degree of generalization and feature leakage.

The encoder $h_\theta$ and **SEAL** decoder $g_\omega$ are jointly optimized under the objective in eq. (**SEAL**). Our method eq. (MQMI) is compared against several baselines that are all based on kernel density estimation (KDE), including divergence-based measures, such as MMD (Gretton et al., 2012), CS divergence (Jenssen, 2024; Jenssen et al., 2006), and a mutual information-based criterion (CS-QMI) (Yu et al., 2024). These methods form a natural basis for comparison under hyperspherical distribution: MMD measures the distance between means of two distributions after mapping them into a Reproducing Kernel Hilbert Space, CS divergence measures the log of angular similarity, and CS-QMI extends CS divergence to estimate quadratic mutual information.

We assess disruption performance across two key tasks: **(i)** Classification: We evaluate both linear and nonlinear inference attacks by training a logistic regression classifier and a simple multi-layer perceptron (MLP) on perturbed embeddings and evaluating on clean embeddings, in order to assess the impact of perturbations on decision boundaries; **(ii)** Retrieval: We examine the effect of perturbations on $k$-nearest-neighbor ($k$-NN) retrieval accuracy, which reflects the preservation or destruction of local geometry in the embedding space.

Table 1 summarizes the results. MQMI achieves strong feature disruption across datasets, particularly on vision tasks, where it often outperforms divergence-based methods (MMD, CS divergence) as well as the mutual information-based CS-QMI. This trend is also broadly reflected in the nonlinear MLP evaluation, suggesting that the disruption effect is not limited to linear decision boundaries. In several cases, the feature leakage drops below random guess levels, indicating that the perturbations induce substantial distributional shifts.

While the improvements are more marked on vision datasets, the gains on NLP tasks are comparatively more modest and less consistent. In particular, although MQMI performs strongly on 20 Newsgroups, its advantage is less consistent on AG News, where CS-Div achieves stronger reductions across multiple metrics. This suggests that the effectiveness of disruption may depend on the underlying embedding space, which can vary across datasets and backbone models.

We observe that CS divergence consistently outperforms MMD in this experiment setting across both vision and NLP tasks. While both methods rely on kernel mean embeddings, CS divergence uses a logarithmic measure of angular similarity Jenssen et al. (2006), whereas MMD measures Euclidean distance.

## 4.2 Privacy Evaluation

We evaluate the effectiveness of disruption in protecting against privacy risks, including membership inference (M-Acc) (Hu et al., 2022; Shokri et al., 2017), where the attacker tries to distinguish training samples from unseen ones, and attribute inference (A-Acc) attacks (Jia & Gong, 2018), which aims to recover sensitive labels from embeddings. For vision datasets, UTKFace evaluates gender ($A_G$) and ethnicity ($A_E$), while CelebA-Smiles evaluates attractiveness ($A_A$) and gender ($A_M$). For text datasets (20 Newsgroup and AG News), we assess general attribute prediction attacks. We report accuracy, precision, and AUC, where lower values indicate stronger privacy preservation.

In each attack, a linear classifier is trained on perturbed embeddings to predict membership or attribute labels. For vision data, classifiers are trained on perturbed embeddings extracted by ViT-Base (DINO) models; for text data, BERT-base embeddings are used. All evaluations are conducted using clean embeddings at test time to simulate realistic exploitation scenarios.

Table 2 summarizes the results. On vision datasets, MQMI achieves near-random performance in membership inference attacks (e.g., around 50%), and provides notable improvements in attribute inference attacks, particularly for gender and attractiveness prediction, where substantial reductions are observed compared to baseline methods. However, the gains are not uniform across all metrics; for example, in some membership inference metrics MQMI is comparable to or slightly worse than CS-Div.

In contrast, the improvements on text datasets are more mixed. While MQMI achieves competitive or improved performance in certain metrics (e.g., attribute accuracy and precision on 20 Newsgroups), its advantage over CS-Div and other baselines is less consistent, especially for membership inference, where MQMI often does not improve over CS-Div and in some cases performs worse.

Overall, these results suggest that MQMI can effectively reduce privacy risks under the evaluated setting, with comparatively stronger gains on vision representations, while exhibiting more modest, non-uniform, and dataset-dependent behavior on text embeddings.

## 4.3 Ablation Studies

We conduct ablation studies to systematically characterize the behavior and sensitivity of **SEAL** with respect to key hyperparameters and architectural components. Together, these experiments illustrate how different design choices control the latent disruption and reconstruction quality under the evaluated threat model.

### 4.3.1 Effect of $\beta$: Balancing Entropy and Disruption

The hyperparameter $\beta$ controls the balance between entropy regularization and latent disruption in the MQMI surrogate. As shown in Table 3, decreasing $\beta$ generally increases the degree of perturbation, leading to reduced effectiveness of classification and retrieval attacks under our evaluation setting.

On STL-10, decreasing $\beta$ from 0.0 to $-1.0$ substantially reduces R@1 from 41.10% to 1.34% and F1@5 from 46.98% to 1.88%, well below the random baseline (R@1 $\approx$ 10%), indicating strong disruption. On Emotion, a similar reduction is observed in classification accuracy (from 57.85% to 12.80%), although retrieval metrics exhibit less consistent behavior, suggesting a more dataset-dependent response.

Compared with CS-QMI and CS-Div, MQMI provides stronger and more tunable disruption overall, while allowing flexible control over the perturbation strength through $\beta$. UMAP visualizations in Figure 3 further show that decreasing $\beta$ increases dispersion in both the perturbations $\Delta$ and the perturbed embeddings $Z'$, supporting this interpretation.

### 4.3.2 Effect of $\gamma$: Balancing Disruption and Reconstruction

The hyperparameter $\gamma$ controls the relative weight between MQMI and the reconstruction loss. As shown in Table 4, varying $\gamma$ affects the trade-off between disruption and reconstruction, but does not lead to a strictly monotonic trend across metrics.

In particular, smaller $\gamma$ values tend to increase disruption (lower LR), while retrieval performance (e.g., R@5) varies only moderately across the explored range of $\gamma$. This suggests that the reconstruction term can be adjusted without substantially degrading retrieval performance under the evaluated setting.

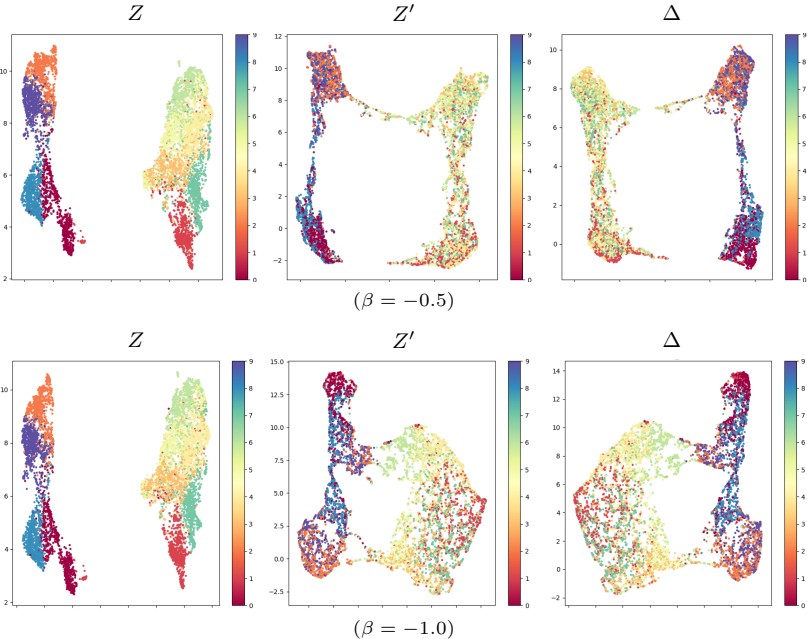

Figure 3: **UMAP Visualizations under Different $\beta$ Values.** Each group shows clean embeddings $Z$, perturbed embeddings $Z'$, and perturbations $\Delta$. Lower $\beta$ leads to stronger disruption.

### 4.3.3 SEAL Decoder vs. Public Decoder

We investigate whether perturbed embeddings can be reconstructed by a public decoder or by our **SEAL** decoder. Both decoders use standard architectures adapted to each dataset (see Appendix B), with token choices discussed in Appendix C.

For vision datasets, as shown in Figure 4, the **SEAL** decoder successfully reconstructs high-quality outputs, while the public decoder fails to recover faithful images from perturbed embeddings. Quantitative evaluations in Table 5 further confirm that only the **SEAL** decoder achieves satisfactory scores across LPIPS (Zhang et al., 2018), SSIM (Wang et al., 2004), PSNR (Horé & Ziou, 2010), and DINO-Sim metrics (Caron et al., 2021b). They suggest that the reconstructed data retains sufficient semantic information for interpretation.

For text datasets, as shown in Table 6, the **SEAL** decoder can reliably reconstruct coherent sentences from perturbed embeddings, whereas the public decoder fails to do so. These results indicate that, under the evaluated setting, meaningful reconstruction from perturbed embeddings is achieved by the **SEAL** decoder.

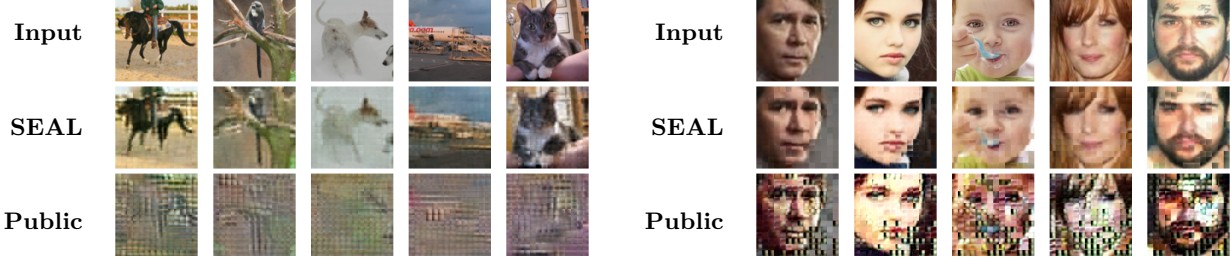

Figure 4: **Public vs. SEAL Reconstructions.** STL-10 and UTKFace examples. From perturbed embeddings, **SEAL** decoder recovers outputs comparable to the inputs, while public decoder fails.

| Dataset | LPIPS ↓ | SSIM ↑ | PSNR ↑ | DINO-Sim ↑ |
|---|---|---|---|---|
| UTKFace | 0.40 | 0.79 | 25.01 | 0.77 |
| Tiny-ImageNet | 0.35 | 0.71 | 22.42 | 0.70 |

Table 5: **SEAL Reconstruction Quality.** Evaluation on vision datasets. Higher values are better for SSIM, PSNR, and DINO-Sim (based on DINO feature similarity); lower is better for LPIPS.

## 5 Conclusion

In this work, we presented **SEAL**, a framework that learns to introduce controlled perturbations in latent embeddings to empirically reduce their misuse under embedding leakage, while retaining sufficient information to support controlled reconstruction for users with access under the considered threat model. The proposed approach formulates disruption through a surrogate objective based on the *Matrix Norm-based Quadratic Mutual Information* (MQMI), providing a computationally efficient mechanism for perturbing latent embeddings.

Through experiments on both vision and text benchmarks, we showed that **SEAL** reduces the effectiveness of evaluated scenarios, including feature leakage and inference attacks, under the considered threat model, while reconstruction remains effective for authorized users in our evaluated setting. Our results highlight an empirical trade-off between limiting latent exploitation and supporting human interpretation through reconstruction, and position **SEAL** as a step toward understanding and mitigating risks associated with sharing latent embeddings.

**Limitations and Future Work**    While **SEAL** demonstrates empirical effectiveness in reducing the misuse of latent embeddings under the considered threat model, it does not provide formal security guarantees and should not be interpreted as preventing misuse in stronger or unmodeled settings. Several limitations remain.

First, the current framework focuses on preserving overall reconstruction quality for authorized users, but does not explicitly control which semantic attributes are retained. In practice, this may be insufficient for applications that require selective or fine-grained preservation (e.g., local structures in medical data). Extending the decoder with more expressive architectures or incorporating task-aware objectives (e.g., perceptual or region-specific losses) could enable more controlled reconstruction. We leave such selective or application-specific designs for future work.

Second, our threat model is tailored to human-in-the-loop scenarios under access control, where utility is defined through preserving semantically meaningful information for authorized interpretation, coupled with a decoder-based access mechanism. This differs from standard embedding-sharing workflows that prioritize

---

**Original Text**

(1) are you ( two ) joking? is the entire internet flaming you ( two )? ahh!, now i remember that ohmite company was the first introducing " the pink colored resistor ", only for electronics working females ; - )

(2) s african tv in beheading blunder public broadcaster sabc apologises after news bulletin shows footage of american beheaded in iraq.

---

**Reconstruction with Public Decoder**

(1) can are you ( two ) joking? is the entire internet flaming you ( two )? ahh!, now i remember that ohmite company was the first introduced " the pink colored resistor ", only for electronics working female ; - )met met met met met met met met met met met met met met met met met met met met met met met met met met met met met met met met met met met met met met met met met met met met met met met met met met met met met met met met met met met met met met met met

(2) council s african tv in beheading blunder public broadcaster sabc apologises after news bulletin shows footage of american beheaded in iraq. gag gag gag gag gag gag gag gag gag gag gag gag gag gag gag gag gag gag gag gag gag gag gag gag

---

**Reconstruction with SEAL Decoder (Ours)**

(1) are you ( two ) joking? is the entire internet flaming you ( two )? ahh!, now i remember that ohmite company was the first introduced " the pink colored resistor ", only for electronics working women ; - )

(2) s african tv in beheading blunder public broadcaster sabc apologises after news bulletin shows footage of american beheaded in iraq. rap rap rap

---

Table 6: **Text reconstruction from perturbed embeddings.** One example from each text dataset is shown, reconstructed by a public decoder and our **SEAL** decoder. The discrete nature of text enables accurate recovery, making the outputs potentially reusable for future applications.

downstream model training on shared representations. Our formulation can be viewed as a conceptual instance of binding utility with access control, where interpretability is preserved for trusted users while reducing misuse risks under exposure. Extending this framework to learning-oriented embedding-sharing pipelines remains an interesting direction for future work.

Third, our threat model assumes that the perturbed embeddings and the **SEAL** decoder are accessible only to trusted parties, and that adversaries do not have access to paired $(X, Z')$ data or the **SEAL** decoder itself. This excludes stronger or adaptive adversaries who may leverage additional resources (e.g., partial supervision, auxiliary data, or decoder/query access) to improve reconstruction or inference. If such access were available, a malicious user could train a surrogate decoder to recover sensitive information, thereby undermining the intended protection. In practice, this corresponds to deployments where data and model access are separated across trust boundaries, and addressing settings without such separation would require additional mechanisms beyond the scope of this work.

Fourth, the use of a **SEAL** decoder introduces an access control mechanism for controlled reconstruction by authorized users. While this can help limit misuse, it also inherently introduces a form of control over information access, raising governance considerations such as who controls decoder access, how authorization is managed, and how reconstructed data are handled. In particular, reconstructed outputs may reintroduce sensitive information if exposed, and therefore require appropriate operational safeguards. These aspects lie beyond the learning objective and are better addressed through suitable governance and security practices.

Fifth, our current evaluation is conducted under relatively controlled adversarial settings, including linear classifiers and a standard public-decoder baseline, and thus does not fully capture stronger adversaries with larger modeling capacity, such as nonlinear inference models, more expressive decoders, or alternative training objectives. Adversaries with additional resources, including larger architectures, increased tuning budgets, or access to auxiliary data, may further enhance their ability to extract information from shared embeddings. Extending the evaluation to such settings would provide a more comprehensive assessment of robustness and remains an important direction for future work.

Finally, in practical deployment scenarios where data distributions evolve over time, the perturbation mechanism (the **SEAL** encoder) and the **SEAL** decoder may require periodic updates or adaptation. Since both components are learned with respect to the underlying embedding distribution, distribution shifts may affect the effectiveness of disruption and reconstruction. In such cases, the framework can be updated through retraining or fine-tuning on newly observed data, with updated components provisioned to trusted parties as needed. This reflects a general requirement for maintaining representation-level protection mechanisms in dynamic environments.

### Acknowledgments

This work was supported by the Research Council of Norway (RCN) through FRIPRO Grant under project number 356103 and its Centres of Excellence scheme, Integreat – Norwegian Centre for knowledge-driven machine learning under project number 332645. This work was partially funded by the RCN under grant no. 309439 and the RCN–NRF (National Research Foundation of Korea) joint project (359216, RS-2025-03522980).

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

## Appendix

## A   Datasets

We evaluate our method on a diverse collection of vision and language datasets to assess generalizability, robustness, and privacy-preserving capabilities across domains.

**STL-10:** A dataset designed for unsupervised and semi-supervised learning, containing 5,000 labeled training images and 8,000 test images across 10 visual categories. All images are resized to $224 \times 224$ and normalized using standard ImageNet statistics (mean: $[0.485, 0.456, 0.406]$, std: $[0.229, 0.224, 0.225]$).

**Tiny-ImageNet:** A subset of ImageNet comprising 200 object categories, with each category containing 500 training images, 50 validation images, and 50 test images. Images are resized to $224 \times 224$ and normalized identically to STL-10.

**CelebA-Smiles:** A subset of the CelebA dataset curated for attribute prediction tasks. It contains approximately 50,000 aligned facial images, annotated with multiple binary attributes. We focus on two attributes: *gender* (male/female) and *attractiveness* (attractive/not attractive). All images are resized to $224 \times 224$.

**UTKFace:** A large-scale facial dataset annotated with demographic attributes including *age*, *gender* (male/female), and *ethnicity* (White, Black, Asian, Indian, Others). The dataset consists of approximately 23,700 images, all resized to $224 \times 224$.

**AG News:** A large-scale text classification benchmark consisting of 120,000 training and 7,600 test news articles across four topics: *World*, *Sports*, *Business*, and *Sci/Tech*. We use the SetFit/HuggingFace version with standard BERT tokenization.

---

Example from `AG News`

Text: "The stock market continued its rally today, buoyed by strong earnings reports from tech companies."
Label: *Business*

---

**Emotion:** A crowd-annotated Twitter dataset comprising 20,000 short English texts labeled with one of six emotion categories: *anger*, *fear*, *joy*, *love*, *sadness*, and *surprise*. We use the standard split of 16,000 training and 4,000 test samples, with BERT-style tokenization.

---

Example from `Emotion` Dataset

Text: "I just got promoted to manager. This is amazing!"
Label: *joy*

---

**20 Newsgroups:** A widely used benchmark for topic classification, containing approximately 18,846 newsgroup posts across 20 different categories, such as *comp.graphics*, *sci.space*, and *rec.sport.hockey*. We adopt the HuggingFace SetFit version with 11,314 training and 7,532 test samples. Standard BERT-style tokenization is applied.

---

Example from 20 `Newsgroups`

Text: "NASA has scheduled the next shuttle launch for June. Looking forward to another milestone in space exploration."
Label: *sci.space*

---

For all image datasets, we resize inputs to $224 \times 224$ and apply consistent normalization. For all text datasets, we use BERT tokenization without additional preprocessing. We retain the original train-test splits for all datasets to ensure fair comparison and reproducibility.

# B    Experimental Setup

In this section, we detail the experimental setup, including datasets, models, hyperparameters, and evaluation metrics used in our study.

## B.1    Preprocessing

We conduct experiments on both computer vision and natural language processing (NLP) tasks:

- **STL-10:** All images are resized to $224 \times 224$ for compatibility with image encoders. We use an 80/20 train-validation split and keep the official test set unchanged.

- **Tiny-ImageNet:** Images originally in $64 \times 64$ resolution are resized to $224 \times 224$. We follow the official train-test split protocol.

- **CelebA-Smiles:** We focus on two binary attributes, *gender* (male/female) and *attractiveness* (attractive/not attractive). All images are resized to $224 \times 224$.

- **UTKFace:** We evaluate privacy inference on two attributes: *gender* (male/female) and *ethnicity* (five categories). All images are resized to $224 \times 224$.

- **Emotion and 20 Newsgroups:** Text data is tokenized using standard BERT tokenization, and sentence embeddings are derived from pre-trained transformers.

## B.2    Models and Implementation Details

We use the following encoder-decoder architectures:

- **Encoders:**
    - **MAE** (He et al., 2021) and **DINO** (Caron et al., 2021b): Used for vision datasets (STL-10, Tiny-ImageNet, CelebA-Smiles, UTKFace).
    - **BERT** (Devlin et al., 2019): Used for text datasets (Emotion, 20 Newsgroups, AG News).

- **Decoder:**
    - For vision datasets, we use a ViT-style decoder with 6 or 8 layers, 16 heads, and an embedding dimension of 512.
    - For text datasets, the decoder consists of 4 or 6 transformer layers with 8 heads, a hidden size of 768, and a feedforward dimension of 1,024.

- **Perturbation Encoder:** A shallow transformer encoder $h_\theta$ with 1 layer, 16 heads, embed dimension 768, and MLP ratio 4.0, trained to perturb embeddings $Z \to Z'$ under different discrepancy objectives, including MQMI (ours), MMD (Gretton et al., 2012), CS divergence (Jenssen, 2024; Jenssen et al., 2006), and CS-QMI (Yu et al., 2024).

## B.3    Hyperparameters

Unless otherwise noted, the following configurations are used throughout all experiments.

### B.3.1    Training Epochs

We train for:

- 20 epochs on STL-10, Emotion, and 20 Newsgroups.

- 5 epochs on Tiny-ImageNet.

- 10 epochs on CelebA-Smiles and UTKFace.

### B.3.2 Learning Rate

We set the learning rate to $5 \times 10^{-4}$ for both the **SEAL** decoder and the perturbation encoder.

### B.3.3 Key Hyperparameters

- $\beta$: Weight of the structural perturbation term $\mathbf{S}_2(\Delta)$.

- $\gamma$: Weight of the reconstruction loss.

- $\kappa$: Hyperparameter of the von Mises–Fisher (vMF) kernel.

## B.4 Evaluation Metrics

We adopt the following evaluation protocols:

- **Linear Classification (LR):** Measures whether a simple logistic regression can separate classes from embeddings.

- **Nearest Neighbor Retrieval:** Recall@1, Recall@5, and F1@5 computed using $k$-NN search.

- **Clustering Quality:** Normalized Mutual Information (NMI) between predicted clusters and ground truth labels.

- **Reconstruction:** We evaluate the reconstruction quality of decoders from perturbed embeddings using both visual inspection (for images and texts) and quantitative metrics. For image datasets (e.g., UTKFace, Tiny-ImageNet), we report LPIPS (lower is better), SSIM (higher is better), PSNR (higher is better), and patch-level similarity (PatchSim, higher is better). For text datasets, qualitative comparisons of generated texts are provided to assess semantic preservation.

- **Membership Inference Attack (MIA):** Using per-sample negative log-likelihood (NLL) from linear classifiers to distinguish training vs. test embeddings. Metrics include attack accuracy (M-Acc), precision (M-Prec), and AUC (M-AUC).

- **Attribute Inference Attack (AIA):** Evaluates attribute classification accuracy, precision, and AUC from perturbed embeddings.

## B.5 Hardware

All experiments are conducted on a single NVIDIA A100 or 3090 GPU within a cluster environment. The computational cost remains reasonable, as only the SEAL encoder and a standard **SEAL** decoder are jointly trained, while the pretrained backbone encoders are kept frozen throughout training.

# C   Study of Inner Product Forms and Token Selection

In our framework, the MQMI functional is primarily applied to the CLS token, but it can also be extended, more aggressively to both the CLS token and the averaged patch tokens (as in ViT). This allows us to disrupt downstream task performance while maintaining reconstruction quality. The design ensures that only the jointly trained **SEAL** decoder can reliably recover the original input.

To examine the influence of different token configurations and inner product forms, we explore three variants for our MQMI objective:

- **Method A**: MQMI is applied separately to the class token and the averaged patch tokens.

- **Method B**: MQMI is applied separately to the class token and all individual patch tokens, using the matrix inner product defined as $\langle A, B \rangle = \text{trace}(A^\top B)$ for $A, B \in \mathbb{R}^{m \times n}$, where $m$ denotes the number of patch tokens and $n$ is the latent feature dimension. The resulting similarity values are further normalized by the Frobenius norm.

- **Method C**: MQMI is applied jointly to all tokens (class and patch) as a single matrix input, i.e., $A, B \in \mathbb{R}^{(m+1) \times n}$, where $m + 1$ corresponds to the total number of tokens including the class token.

We quantitatively evaluate these three configurations on STL-10 using standard feature leakage metrics. As shown in Table 7, Method B achieves the lowest scores across all metrics, indicating a stronger disruption of the learned representations. Method C, however, provides minimal improvement over the baseline, suggesting that a joint estimator across all tokens might fail to effectively isolate and suppress sensitive components.

| Method | LR ↓ | R@1 ↓ | R@5 ↓ | F1@5 ↓ | NMI ↓ |
|---|---|---|---|---|---|
| Baseline | 99.12 | 98.68 | 99.66 | 98.90 | 93.96 |
| Method A | 54.93 | 93.13 | 99.38 | 94.13 | 16.53 |
| Method B | **43.49** | **89.86** | **99.16** | **91.29** | **12.76** |
| Method C | 99.06 | 98.62 | 99.65 | 98.88 | 93.18 |

Table 7: **Ablation on MQMI application strategies (STL-10)**. Method B demonstrates the strongest disruption across all feature leakage metrics.

To visually illustrate these quantitative results, we compare reconstruction outputs from the **SEAL** and public decoders in each setting. All models are trained using the same DINO encoder on STL-10, and the reconstructions are presented in Figure 5.

In summary, while Method B achieves the strongest disruption both quantitatively and visually, it introduces additional complexity due to matrix-level normalization. In contrast, Method A provides a more heuristic and lightweight alternative, yet still delivers effective disruption and satisfactory reconstruction performance. Considering this trade-off and its strong results in our main experiments, we adopt Method A and apply MQMI to CLS tokens as the default configuration throughout the paper.

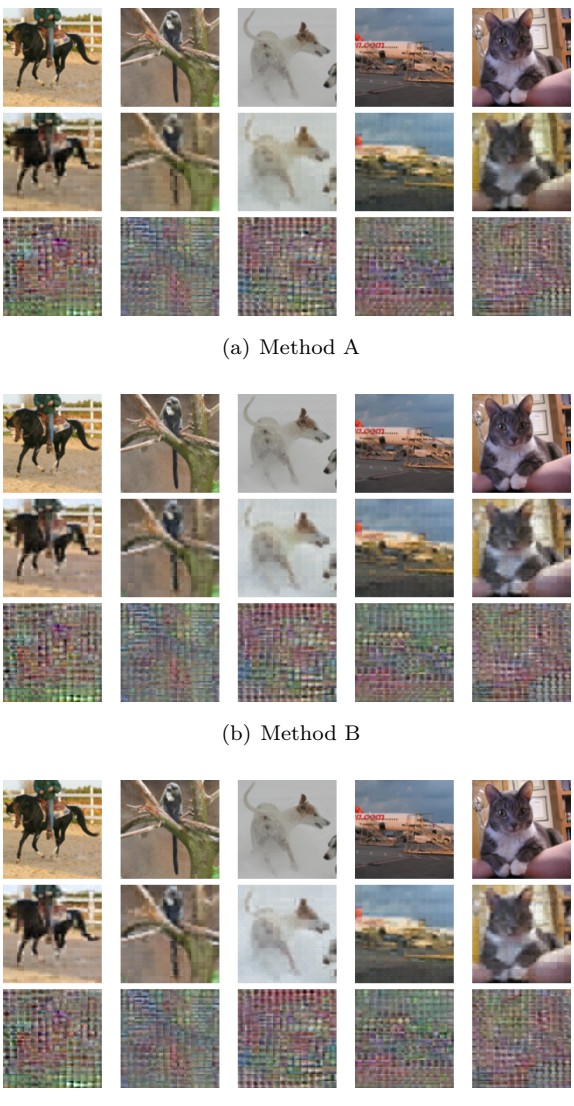

(a) Method A

(b) Method B

(c) Method C

Figure 5: Visual comparison between reconstructions from the **SEAL** decoder (second row) and the public decoder (third row) under three MQMI application strategies (Methods A, B, and C) on STL-10 using a DINO encoder. All three methods show clear qualitative differences between **SEAL** and public reconstructions, confirming the strong disruptive capability of MQMI and the uniqueness of the **SEAL** decoder, which together constitute the core idea of our **SEAL** framework.

# D  Geometric Motivation

This section provides a brief geometric intuition for using $\mathbf{S}_2(\Delta)$ in our surrogate for controlled perturbations.

We work under a simple high-dimensional spherical perturbation model: (i) the samples $z_1, \dots, z_N$ are i.i.d. draws from a von Mises–Fisher distribution on $\mathbb{S}^{d-1}$; (ii) conditionally on the realized samples, the perturbations $\delta_1, \dots, \delta_N$ are independent across $i$; and (iii) the tangent components of these perturbations are conditionally centered, isotropic on the corresponding tangent spaces, and satisfy a concentration condition. Under this model, the derivation is intended to explain why the perturbation contribution can be separated from the original similarity structure and summarized through $\mathbf{S}_2(\Delta)$, rather than to establish a strict closed-form equivalence.

Building on the geometric decomposition above, we analyze how the perturbation propagates from inner products to kernel and entropy functionals. The perturbed kernel can be expressed as a product of an original similarity term, a perturbation-induced term, and a residual factor. This yields an approximate factorization of the normalized Frobenius norm, leading to a decomposition of $\mathbf{S}_2(Z')$ into contributions from $\mathbf{S}_2(Z)$, $\mathbf{S}_2(\Delta)$, and lower-order residual terms. We emphasize that this factorization is not exact, but holds up to approximation and finite-sample errors. We further analyze the perturbation kernel on $\Delta$ in relation to the standard RBF parameterization. At the pairwise level, the perturbation term admits a representation as a power-transformed kernel, with the coefficient $\beta(\rho)$ arising from aligning the scale induced by the vMF expansion with the RBF bandwidth. When passing to entropy functionals, this correspondence no longer holds exactly; instead, under an additional concentration assumption, it leads to a surrogate approximation that relates $\mathbf{S}_2(\Delta)$ to the perturbation contribution in the entropy functional.

## D.1  vMF Kernel and Spherical Geometry

Let $z_1, \dots, z_N \in \mathbb{S}^{d-1}$ be i.i.d. samples from a von Mises–Fisher (vMF) distribution with density

$$p(z) = C_d(\kappa) \exp(\kappa\, z^\top \mu), \qquad \|\mu\|_2 = 1,\; \kappa \geq 0.$$

The corresponding vMF kernel is

$$\kappa_{\mathrm{vMF}}(u, v) = \exp(\kappa\, u^\top v), \qquad u, v \in \mathbb{S}^{d-1},$$

which depends only on the angle $\theta$ between $u$ and $v$, since $u^\top v = \cos\theta$. For $u, v \in \mathbb{S}^{d-1}$,

$$\|u - v\|_2^2 = 2(1 - u^\top v) = 2(1 - \cos\theta).$$

## D.2  Perturbations on the Sphere

Consider perturbed samples $z_i' = z_i + \delta_i \in \mathbb{S}^{d-1}$, where each $\delta_i \in \mathbb{R}^d$ is an arbitrary chord vector. The unit–norm constraint gives the exact identity

$$1 = \|z_i'\|_2^2 = 1 + 2\, z_i^\top \delta_i + \|\delta_i\|_2^2 \implies z_i^\top \delta_i = -\tfrac{1}{2} \|\delta_i\|_2^2,$$

which holds for any $\delta_i$ that places $z_i'$ on the unit sphere.

Conditionally on the realized samples $\{z_i\}_{i=1}^N$, let $\delta_1, \dots, \delta_N$ be sample-wise perturbations satisfying $z_i' = z_i + \delta_i \in \mathbb{S}^{d-1}$ for each $i$, and assume that $\delta_1, \dots, \delta_N$ are independent. Their tangent components

$$\delta_i^\perp := \delta_i - (z_i^\top \delta_i) z_i$$

are conditionally isotropic on the tangent space $T_{z_i}\mathbb{S}^{d-1} = \{v \in \mathbb{R}^d : z_i^\top v = 0\}$, in the sense that

$$\mathbb{E}\big[\, \delta_i^\perp (\delta_i^\perp)^\top \mid z_i \big] = \frac{\tau^2}{d - 1}\, (I_d - z_i z_i^\top), \qquad \|\delta_i^\perp\|_2^2 = \tau^2 + O_p(d^{-1/2}),$$

with sub-Gaussian concentration on the tangent space. Here $I_d - z_i z_i^\top$ is the orthogonal projector onto $T_{z_i}\mathbb{S}^{d-1}$, so the isotropy is understood within the tangent subspace rather than in the ambient Euclidean basis. The radial component is then fully determined by the spherical constraint above, giving

$$z_i^\top \delta_i = -\tfrac{1}{2}\|\delta_i\|_2^2 = -\tfrac{1}{2}\tau^2 + O_p(d^{-1/2}).$$

For $i \neq j$, we decompose the cross-term as

$$z_i^\top \delta_j = z_i^\top \delta_j^\perp + (z_j^\top \delta_j)\, z_i^\top z_j.$$

Conditionally on $z_i$ and $z_j$, the tangent component $\delta_j^\perp$ is centered, and its conditional covariance is isotropic on the tangent space $T_{z_j}\mathbb{S}^{d-1}$. Therefore,

$$\mathbb{E}\big[z_i^\top \delta_j^\perp \mid z_i, z_j\big] = z_i^\top \mathbb{E}\big[\delta_j^\perp \mid z_i, z_j\big] = 0.$$

Moreover, using the conditional covariance assumption,

$$\mathrm{Var}\big(z_i^\top \delta_j^\perp \mid z_i, z_j\big) = z_i^\top \mathbb{E}\big[\delta_j^\perp (\delta_j^\perp)^\top \mid z_i, z_j\big] z_i = z_i^\top \frac{\tau^2}{d-1}(I_d - z_j z_j^\top) z_i.$$

Since $\|z_i\|_2 = 1$, this simplifies to

$$\mathrm{Var}\big(z_i^\top \delta_j^\perp \mid z_i, z_j\big) = \frac{\tau^2}{d-1}\big(1 - (z_i^\top z_j)^2\big) \leq \frac{\tau^2}{d-1} = O(d^{-1}).$$

By the assumed concentration of the tangent perturbations, this variance scaling implies

$$z_i^\top \delta_j^\perp = O_p(d^{-1/2}).$$

For the radial correction term, the spherical constraint gives

$$z_j^\top \delta_j = -\tfrac{1}{2}\|\delta_j\|_2^2 = -\tfrac{1}{2}\tau^2 + O_p(d^{-1/2}).$$

In addition, for independent vMF samples in high dimension, the pairwise inner products satisfy

$$z_i^\top z_j = O_p(d^{-1/2}).$$

Hence

$$(z_j^\top \delta_j)\, z_i^\top z_j = O_p(d^{-1/2}).$$

Combining the tangent and radial contributions yields

$$z_i^\top \delta_j = O_p(d^{-1/2}), \qquad z_j^\top \delta_i = O_p(d^{-1/2}).$$

For $i = j$, the projection $\langle z_i, \delta_i \rangle$ is concentrated around $-\tfrac{1}{2}\tau^2$ with fluctuations of order $O_p(d^{-1/2})$, consistent with the concentration behavior of spherical chords; this self-case isotropy is verified empirically in Figure 6. For $i \neq j$, the cross-terms have conditional mean zero up to the negligible radial correction above, and variance of order $O(d^{-1})$, as confirmed empirically in Figure 7.

### D.3  Inner-product Expansion

Starting from $z_i' = z_i + \delta_i$, we expand the perturbed inner product:

$$z_i'^\top z_j' = z_i^\top z_j + z_i^\top \delta_j + \delta_i^\top z_j + \delta_i^\top \delta_j. \tag{1}$$

Using $z_i^\top \delta_i = -\frac{1}{2}\|\delta_i\|^2$ and $z_j^\top \delta_j = -\frac{1}{2}\|\delta_j\|^2$, the perturbed inner product admits the decomposition

$$z_i'^\top z_j' = z_i^\top z_j - \tfrac{1}{2}\|\delta_i - \delta_j\|_2^2 + r_{ij}, \tag{Decomp}$$

where $r_{ij} := z_i^\top(\delta_j - \delta_i) + z_j^\top(\delta_i - \delta_j)$ collects all higher–order and cross terms, and satisfies $r_{ij} = O(1) + O_p(d^{-1/2})$ under the isotropy assumptions.

Equation (Decomp) shows that the perturbed inner product splits into three components: the original vMF interaction $z_i^\top z_j$, the chord-induced Euclidean term $\|\delta_i - \delta_j\|_2^2$ that yields the Gaussian factor, and a bounded residual $r_{ij}$ contributing only a constant multiplier inside the kernel. This decomposition is precisely what enables the near-additive factorization of the normalized kernel.

### D.4 Kernel Factorization

For the vMF kernel

$$\kappa_{\mathrm{vMF}}(u, v) = \exp(\kappa\, u^\top v),$$

the perturbed kernel entries satisfy

$$K_{ij}^{Z'} = \exp\!\Big(\kappa z_i^\top z_j - \tfrac{\kappa}{2}\|\delta_i - \delta_j\|_2^2 + \kappa r_{ij}\Big),$$

where $r_{ij} = O(1) + O_p(d^{-1/2})$ collects the residual inner-product terms from the expansion in the previous section.

To analyze the Frobenius norm, we first square each entry:

$$(K_{ij}^{Z'})^2 = \exp\!\Big(2\kappa z_i^\top z_j - \kappa\|\delta_i - \delta_j\|_2^2 + 2\kappa r_{ij}\Big). \tag{6}$$

We now separate the three contributions appearing in Equation (6). Define

$$f_{ij} := \exp(2\kappa z_i^\top z_j),$$

$$g_{ij} := \exp\!\big(-\kappa\|\delta_i - \delta_j\|_2^2\big),$$

$$h_{ij} := \exp(2\kappa r_{ij}).$$

Then

$$(K_{ij}^{Z'})^2 = f_{ij}\, g_{ij}\, h_{ij}. \tag{7}$$

Substituting Equation (7) into the Frobenius norm gives

$$\|K^{Z'}\|_F^2 = \sum_{i,j}(K_{ij}^{Z'})^2 = \sum_{i,j} f_{ij}g_{ij}h_{ij}.$$

To express the Frobenius norm as an average over i.i.d. samples, let

$$X_i := (z_i, \delta_i),$$

and define the symmetric kernel

$$\psi(X_i, X_j) := f_{ij}g_{ij}h_{ij}.$$

Since $\{X_i\}_{i=1}^N$ are i.i.d. and $\psi$ depends only on the pair $(X_i, X_j)$, the empirical average

$$\frac{1}{N^2}\sum_{i,j}\psi(X_i, X_j)$$

is a second-order V-statistic associated with $\psi$. Standard laws of large numbers for V-statistics therefore yield

$$\frac{1}{N^2} \sum_{i,j} f_{ij} g_{ij} h_{ij} = \mathbb{E}[f_{12} g_{12} h_{12}] + O_p(N^{-1/2}). \tag{8}$$

Next, introduce the empirical averages

$$S_{Z,h}^{(N)} := \frac{1}{N^2} \sum_{i,j} f_{ij} g_{ij} h_{ij},$$

$$S_Z^{(N)} := \frac{1}{N^2} \sum_{i,j} f_{ij}, \qquad S_\Delta^{(N)} := \frac{1}{N^2} \sum_{i,j} g_{ij}.$$

By Equation (8),

$$S_{Z,h}^{(N)} = \mathbb{E}[f_{12} g_{12} h_{12}] + O_p(N^{-1/2}),$$

and applying the same V-statistic law of large numbers to $f_{ij}$ and $g_{ij}$ separately gives

$$S_Z^{(N)} = \mathbb{E}[f_{12}] + O_p(N^{-1/2}),$$

$$S_\Delta^{(N)} = \mathbb{E}[g_{12}] + O_p(N^{-1/2}).$$

We now define the bounded residual factor

$$c_\kappa := \frac{\mathbb{E}[f_{12} g_{12} h_{12}]}{(\mathbb{E}[f_{12}])(\mathbb{E}[g_{12}])}, \qquad c_\kappa = O(1),$$

which collects the remaining effect of the cross-term factor $h_{12}$. In general, this term may still vary with the perturbation distribution and therefore with optimization. In the present derivation, however, we treat this variation as part of the higher-order approximation error, and focus on the leading contribution captured by the separated terms $\mathbb{E}[f_{12}]$ and $\mathbb{E}[g_{12}]$. Thus

$$\mathbb{E}[f_{12} g_{12} h_{12}] = c_\kappa \left(\mathbb{E}[f_{12}]\right)(\mathbb{E}[g_{12}]).$$

Substituting the previous three expansions into this identity yields

$$S_{Z,h}^{(N)} = c_\kappa \, S_Z^{(N)} \, S_\Delta^{(N)} \left(1 + O_p(N^{-1/2})\right).$$

Since

$$S_{Z,h}^{(N)} = \frac{1}{N^2} \sum_{i,j} f_{ij} g_{ij} h_{ij} = \frac{1}{N^2} \|K^{Z'}\|_F^2,$$

we obtain

$$\frac{1}{N^2} \|K^{Z'}\|_F^2 = S_Z^{(N)} S_\Delta^{(N)} c_\kappa \left(1 + O_p(N^{-1/2})\right). \tag{9}$$

To identify the factors $S_Z^{(N)}$ and $S_\Delta^{(N)}$, note that

$$K_{ij}^Z := \exp(\kappa z_i^\top z_j), \qquad K_{ij}^\Delta := \exp\left(-\frac{\kappa}{2} \|\delta_i - \delta_j\|_2^2\right).$$

Then, by construction,

$$f_{ij} = (K_{ij}^Z)^2, \qquad g_{ij} = (K_{ij}^\Delta)^2,$$

and therefore

$$S_Z^{(N)} = \frac{1}{N^2} \|K^Z\|_F^2, \qquad S_\Delta^{(N)} = \frac{1}{N^2} \|K^\Delta\|_F^2.$$

Now let the normalized kernels be defined by

$$A^Z := \frac{K^Z}{\mathrm{tr}(K^Z)}, \qquad A^\Delta := \frac{K^\Delta}{\mathrm{tr}(K^\Delta)}, \qquad A^{Z'} := \frac{K^{Z'}}{\mathrm{tr}(K^{Z'})}.$$

Since $z_i^\top z_i = 1$ and $\delta_i - \delta_i = 0$, the diagonal entries satisfy

$$K_{ii}^Z = e^\kappa, \qquad K_{ii}^{Z'} = e^\kappa, \qquad K_{ii}^\Delta = 1.$$

Hence

$$\mathrm{tr}(K^Z) = Ne^\kappa, \qquad \mathrm{tr}(K^{Z'}) = Ne^\kappa, \qquad \mathrm{tr}(K^\Delta) = N.$$

Therefore

$$\frac{1}{N^2}\|K^Z\|_F^2 = e^{2\kappa}\|A^Z\|_F^2, \qquad \frac{1}{N^2}\|K^\Delta\|_F^2 = \|A^\Delta\|_F^2, \qquad \frac{1}{N^2}\|K^{Z'}\|_F^2 = e^{2\kappa}\|A^{Z'}\|_F^2.$$

Substituting these identities into Equation (9) gives

$$e^{2\kappa}\|A^{Z'}\|_F^2 = c_\kappa\, e^{2\kappa}\|A^Z\|_F^2\,\|A^\Delta\|_F^2\,\big(1 + O_p(N^{-1/2})\big).$$

Canceling the common factor $e^{2\kappa}$ on both sides yields

$$\|A^{Z'}\|_F^2 = c_\kappa\,\|A^Z\|_F^2\,\|A^\Delta\|_F^2\,\big(1 + O_p(N^{-1/2})\big). \tag{10}$$

Taking negative logarithms of Equation (10) gives

$$\mathbf{S}_2(Z') = \mathbf{S}_2(Z) + \mathbf{S}_2(\Delta) + C_\kappa + O_p(N^{-1/2}),$$

where

$$\mathbf{S}_2(Z) := -\log\|A^Z\|_F^2, \qquad \mathbf{S}_2(\Delta) := -\log\|A^\Delta\|_F^2, \qquad C_\kappa := -\log c_\kappa.$$

Here $C_\kappa$ represents a residual (approximation) error induced by the unmodeled interaction terms, whereas the $O_p(N^{-1/2})$ term corresponds to the sample error arising from finite-sample averaging.

At this stage, the bandwidth in $\mathbf{S}_2(\Delta)$ is not introduced through a separate KDE choice, but is inherited from the vMF-based perturbation decomposition; under the standard RBF form $K_{ij}^\Delta = \exp(-\|\delta_i - \delta_j\|_2^2/(2\sigma^2))$, this corresponds to the matched scaling $\sigma^2 = 1/(2\kappa)$.

### D.5 Kernel Bandwidth Alignment and Reparameterization

The perturbation term derived from the vMF expansion and the standard RBF kernel are written under different scaling conventions. In particular, the perturbation factor produced by the vMF expansion is parameterized by $\kappa$, whereas the kernel on $\Delta$ is parameterized by the bandwidth $\sigma^2$. To compare them meaningfully, their scales must first be aligned at the level of pairwise similarities.

From the previous derivation, the perturbation contribution appears as

$$g_{ij} = \exp\big(-\kappa\|\delta_i - \delta_j\|_2^2\big).$$

On the other hand, the standard RBF kernel on the perturbation variables is

$$K_{ij}^\Delta = \exp\Big(-\frac{\|\delta_i - \delta_j\|_2^2}{2\sigma^2}\Big).$$

Matching the exponents gives the exact entrywise identity

$$g_{ij} = \big(K_{ij}^\Delta\big)^{2\kappa\sigma^2}.$$

Introducing the dimensionless parameter

$$\rho := \frac{\kappa\sigma^2}{2}, \qquad \beta(\rho) := 4\rho = 2\kappa\sigma^2,$$

we can rewrite this as

$$g_{ij} = \left(K_{ij}^\Delta\right)^{\beta(\rho)}.$$

This makes the role of $\beta(\rho)$ explicit: it is the exact scaling needed to make the perturbation term derived from the vMF expansion coincide with a power-transformed kernel on $\Delta$. In this sense, $\beta(\rho)$ is not introduced ad hoc; it is the balancing coefficient required to align the two parameterizations. In the matched case $\sigma^2 = 1/(2\kappa)$, one has $\beta(\rho) = 1$, and the equality reduces to $g_{ij} = K_{ij}^\Delta$.

This exact correspondence holds at the pairwise level. The main difficulty arises only when passing from entrywise powers to a global quantity such as a normalized Frobenius norm or entropy functional, since in general

$$\sum_{i,j}\left(K_{ij}^\Delta\right)^{\beta(\rho)} \neq \left(\sum_{i,j}(K_{ij}^\Delta)^2\right)^{\beta(\rho)/2}.$$

Accordingly, the bandwidth effect should not be interpreted as an exact power law on $\|A^\Delta\|_F^2$. Instead, we introduce the exact global functional

$$\Phi_\rho(A^\Delta) := -\log\left(\frac{1}{N^2}\sum_{i,j}\left(K_{ij}^\Delta\right)^{\beta(\rho)}\right),$$

which summarizes the bandwidth-dependent reweighting over all pairs. Under trace normalization, this yields

$$\mathbf{S}_2(Z') \approx \mathbf{S}_2(Z) + \Phi_\rho(A^\Delta) + C_\kappa + O_p(N^{-1/2}),$$

where

$$C_\kappa := -\log c_\kappa.$$

To obtain a tractable surrogate, we impose an additional concentration assumption: the pairwise values $K_{ij}^\Delta$, or equivalently $X_{ij} := -\log K_{ij}^\Delta$, are sufficiently concentrated around a common scale. Under this assumption, changing the exponent mainly rescales the global average rather than changing its overall shape.

Indeed, writing

$$K_{ij}^\Delta = e^{-X_{ij}},$$

we have

$$\Phi_\rho(A^\Delta) = -\log\left(\frac{1}{N^2}\sum_{i,j}e^{-\beta(\rho)X_{ij}}\right), \qquad \mathbf{S}_2(\Delta) = -\log\left(\frac{1}{N^2}\sum_{i,j}e^{-2X_{ij}}\right).$$

When the distribution of $X_{ij}$ is narrow, both expressions are governed by the same typical scale, and a first-order approximation gives

$$\Phi_\rho(A^\Delta) \approx \frac{\beta(\rho)}{2}\mathbf{S}_2(\Delta).$$

Defining the effective global coefficient

$$\tilde\beta(\rho) := \frac{\beta(\rho)}{2} = 2\rho = \kappa\sigma^2,$$

the entropy-level approximation becomes

$$\mathbf{S}_2(Z') \approx \mathbf{S}_2(Z) + \tilde\beta(\rho)\,\mathbf{S}_2(\Delta) + C_\kappa + O_p(N^{-1/2}).$$

Here $C_\kappa$ represents the residual approximation error induced by the unmodeled interaction terms, which may not be negligible in general, whereas the $O_p(N^{-1/2})$ term is the sample error arising from finite-sample averaging. Retaining only the leading-order terms gives the effective objective

$$\mathcal{L} = \mathbf{S}_2(Z) + \tilde\beta(\rho)\,\mathbf{S}_2(\Delta).$$

# E   Information-Theoretic Perspective on Privacy

In this section, we provide a theoretical perspective on how reducing information-theoretic dependence limits the misuse of latent embeddings for sensitive attributes such as class labels.

We begin from Shannon mutual information, which serves as a standard measure of statistical dependence. We then consider the $\alpha$-Rényi entropy family as a generalized framework, where different values of $\alpha$ induce alternative dependence measures.

For practical optimization, we adopt a matrix-based realization of entropy functionals, defined over normalized positive definite (NPD) kernel matrices through their eigenspectrum. Within this formulation, the entropy functional recovers the Shannon differential entropy in the limit $\alpha \to 1$. Under this framework, we use the order-2 case, leading to the matrix-based quadratic mutual information $\mathbf{I}_2(Z; Z')$, which admits a differentiable and computationally efficient formulation.

While this formulation is not equivalent to Shannon mutual information, it provides a tractable and structurally grounded measure of dependence between representations. For theoretical analysis, we adopt the Shannon mutual information framework, which offers well-established tools and guarantees.

**Proposition E.1** (Privacy-Induced Classification Lower Bound). *Let $Y \to X \to Z \to Z'$ be a Markov chain, where $Y \in \mathcal{Y}$ denotes a discrete sensitive attribute, $X$ is the observed input data, $Z$ is the corresponding clean latent representation, and $Z'$ is its perturbed version produced by SEAL. For any classifier $\hat{Y} = f(Z')$ attempting to infer $Y$ from $Z'$, let the misclassification probability be $\varepsilon = \Pr[\hat{Y} \neq Y]$. Then,*

$$\varepsilon \geq \frac{H(Y) - I(Z; Z') - 1}{\log |\mathcal{Y}|}. \tag{11}$$

*Reducing $I(Z; Z')$ therefore raises the lower bound on classification error, implying that weaker mutual dependence between the clean and perturbed embeddings necessarily increases prediction uncertainty and reduces inference capability.*

*Proof.* Consider the Markov chain $Y \to Z \to Z'$, where $Y$ is a discrete label or sensitive attribute, $Z$ the clean latent representation, and $Z'$ its perturbed counterpart produced by MQMI. Let $\hat{Y} = f(Z')$ denote any classifier attempting to predict $Y$ from $Z'$, and define the misclassification probability

$$\varepsilon = \Pr[\hat{Y} \neq Y].$$

From the data-processing inequality,

$$I(Y; Z') \leq I(Z; Z').$$

Using the identity $H(Y|Z') = H(Y) - I(Y; Z')$, we obtain

$$H(Y|Z') \geq H(Y) - I(Z; Z'). \tag{C.1}$$

By Fano's inequality (for $|\mathcal{Y}| \geq 2$),

$$\varepsilon \geq \frac{H(Y|Z') - 1}{\log |\mathcal{Y}|}. \tag{C.2}$$

Combining (C.1) and (C.2) yields

$$\varepsilon \geq \frac{H(Y) - I(Z; Z') - 1}{\log |\mathcal{Y}|},$$

which establishes the result. $\qquad\square$

This result can be interpreted from two perspectives. First, by the data processing inequality, the dependence between the perturbed representation and the sensitive attribute is upper-bounded by that between the clean and perturbed embeddings, i.e., $I(Y; Z') \leq I(Z; Z')$. Second, as $I(Z; Z')$ decreases, Fano's inequality implies that even the Bayes-optimal classifier cannot achieve low prediction error on $Y$ based on $Z'$. Together, these observations indicate that reducing the mutual dependence between $Z$ and $Z'$ limits the amount of information about $Y$ that can be inferred from the secured embedding.

## F   Additional Experiments

### F.1   Empirical Verification of Fixed–Base and Cross–Chord Isotropy

We empirically examine the high–dimensional isotropy properties of spherical chords. For the fixed–base (self) case, we fix a base vector $z = (1, 0, \ldots, 0) \in \mathbb{S}^{d-1}$, and for each dimension $d \in \{128, 256, 512\}$, we uniformly sample $N = 5 \times 10^4$ endpoints $b_i$ on the unit sphere. Each chord is defined as $\delta_i = b_i - z$, and we compute the radial projection $\langle z, \delta_i \rangle$.

Figure 6 shows the resulting histograms. As $d$ increases, the variance of $\langle z, \delta_i \rangle$ shrinks at the expected $O(1/d)$ rate, while the mean remains stable. This confirms that chords emanating from a fixed base point exhibit asymptotically isotropic behavior in the self case $(i = j)$.

For comparison, we also evaluate the cross–chord case $(i \neq j)$, where both $z_i$ and the chord endpoints $b_j$ are independently sampled from $\mathbb{S}^{d-1}$. Figure 7 demonstrates that $\langle z_i, b_j - z_j \rangle$ concentrates around zero with variance scaling as $O(1/d)$, consistent with tangent–space isotropy.

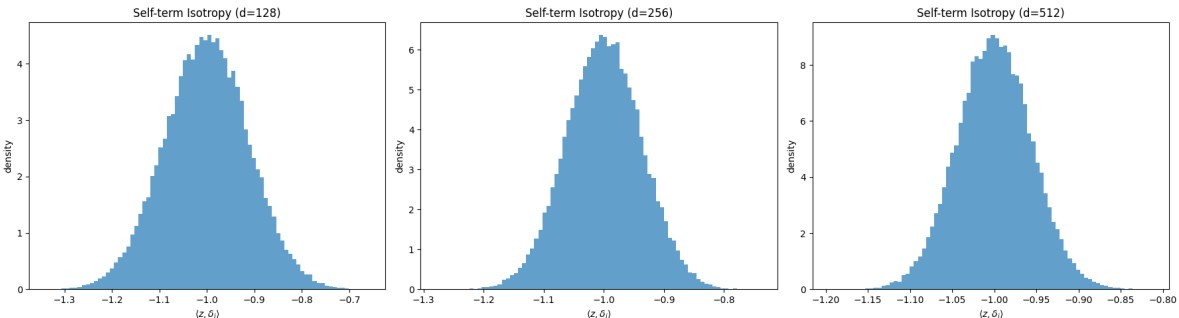

Figure 6: Fixed–base (self) isotropy. Distribution of $\langle z, \delta_i \rangle$ for $d = 128$, $256$, and $512$, where $\delta_i = b_i - z$ and $b_i \sim \text{Unif}(\mathbb{S}^{d-1})$. The variance decreases at rate $O(1/d)$ while the mean remains stable, consistent with high–dimensional spherical chord behavior.

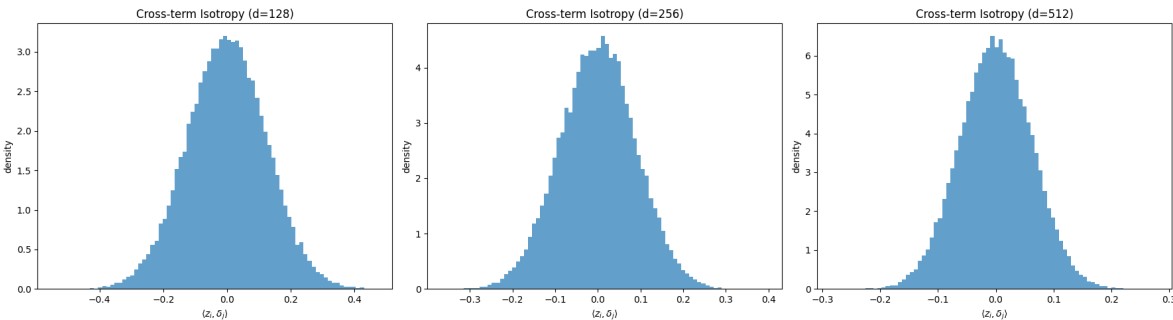

Figure 7: Cross–chord isotropy. Distribution of $\langle z_i, b_j - z_j \rangle$ for $d = 128$, $256$, and $512$, where $z_i$, $z_j$, and $b_j$ are independently sampled from $\text{Unif}(\mathbb{S}^{d-1})$. The distribution concentrates around zero with variance $O(1/d)$, reflecting isotropy of independent spherical chords.

### F.2   Distribution of class tokens

Figure 8 compares the distribution of class token norms across both vision and language datasets. We observe that the class token embeddings in all cases exhibit approximately Gaussian-like patterns, suggesting a natural hyperspherical structure that justifies the use of vMF kernels for modeling.

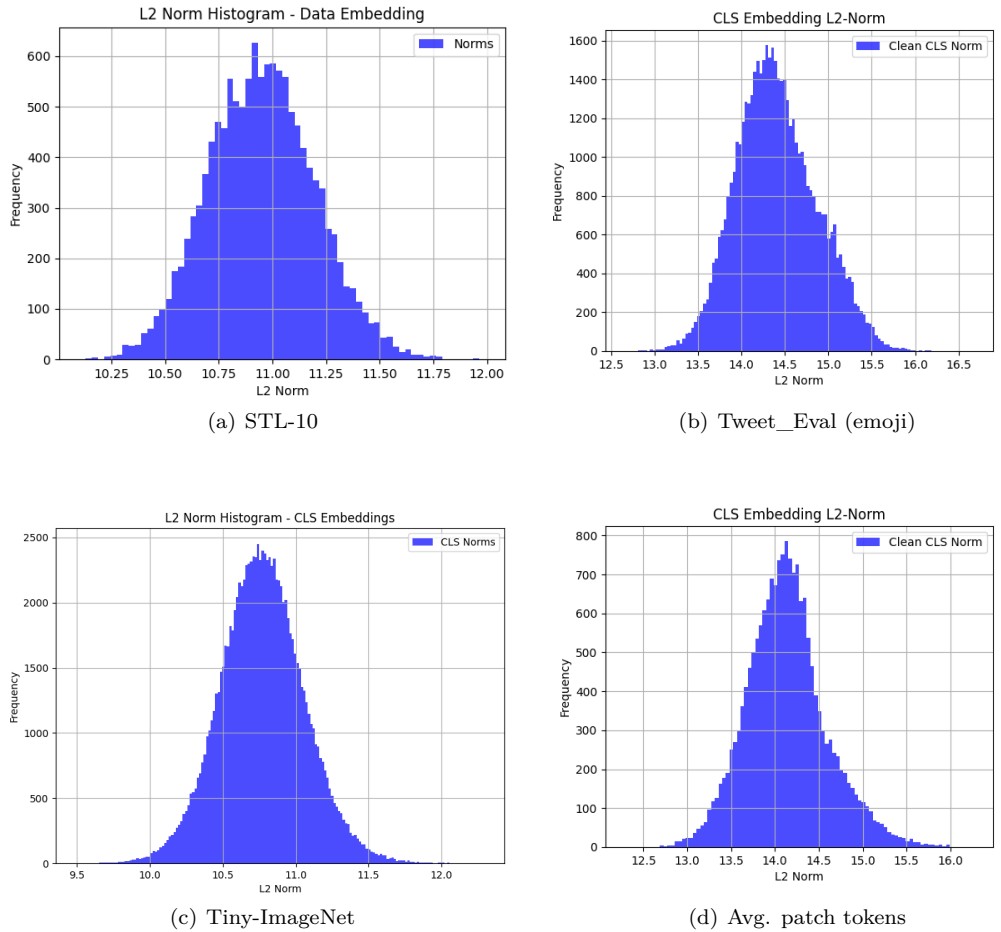

Figure 8: Distribution of class token norms across four datasets. (a) STL-10, (b) Tweet_Eval, (c) Tiny-ImageNet, and (d) Emotion.

| Metric | Baseline | MMD | CS | CS-QMI | MQMI |
|---|---|---|---|---|---|
| LR ↓ | 59.60 | 19.45 | **10.90** | 52.60 | 12.80 |
| R@1 ↓ | 37.30 | 20.80 | **15.90** | 26.30 | 20.00 |
| R@5 ↓ | 79.50 | 72.80 | **60.20** | 72.25 | 66.95 |
| F1@5 ↓ | 50.18 | 34.31 | **27.15** | 39.07 | 30.45 |

Table 8: **Feature Leakage Evaluation on Emotion dataset.** Lower is better (↓).

### F.3   Emotion Dataset

### F.4   Effect of $\kappa$

We study $\kappa$ for the kernel in our MQMI objective. Table 9 presents an ablation on STL-10 and Tiny-ImageNet.

### F.5   Normalized Mutual Information

We use Normalized Mutual Information (NMI) (McDaid et al., 2013) to quantify how much clustering structure aligned with the ground-truth labels remains after perturbation.

| Method | Acc. LR | R@1 | R@5 | F1@5 | NMI |
|---|---|---|---|---|---|
| Baseline | 95.45 | 84.06 | 96.79 | 86.47 | 45.43 |
| $\kappa = 1.0$ | 24.59 | 1.34 | 4.79 | 1.88 | 41.68 |
| $\kappa = 2.0$ | 38.36 | 39.32 | 70.51 | 45.91 | 45.88 |
| $\kappa = 5.0$ | 58.05 | 45.07 | 77.66 | 53.19 | 45.61 |

| Method | Acc. LR | R@1 | R@5 | F1@5 | NMI |
|---|---|---|---|---|---|
| Baseline | 77.78 | 74.62 | 87.84 | 78.16 | 73.03 |
| $\kappa = 0.5$ | – | 72.34 | 86.32 | 76.48 | 66.65 |
| $\kappa = 1.0$ | 73.30 | 68.58 | 85.46 | 73.67 | 56.47 |
| $\kappa = 2.0$ | 68.84 | 69.60 | 85.48 | 74.19 | 56.11 |

Table 9: **Ablation on the kernel hyperparameter $\kappa$.** We vary the concentration parameter $\kappa$ in MQMI. Results are reported for (a) STL-10 and (b) Tiny-ImageNet datasets.

| Dataset | Baseline | MMD | CS-Div | CS-QMI | MQMI |
|---|---|---|---|---|---|
| STL-10 | 45.43 | 47.37 | 47.08 | **39.14** | 41.68 |
| Tiny-ImageNet | 73.03 | 76.13 | 81.21 | **55.22** | 64.78 |
| 20 Newsgroup | 20.73 | **5.81** | 20.07 | 18.34 | 25.65 |
| AG News | 31.36 | **8.38** | 35.37 | 32.97 | 33.60 |

Table 10: **Normalized Mutual Information (NMI).** We report NMI between ground-truth class labels and $k$-means clustering on secured embeddings.

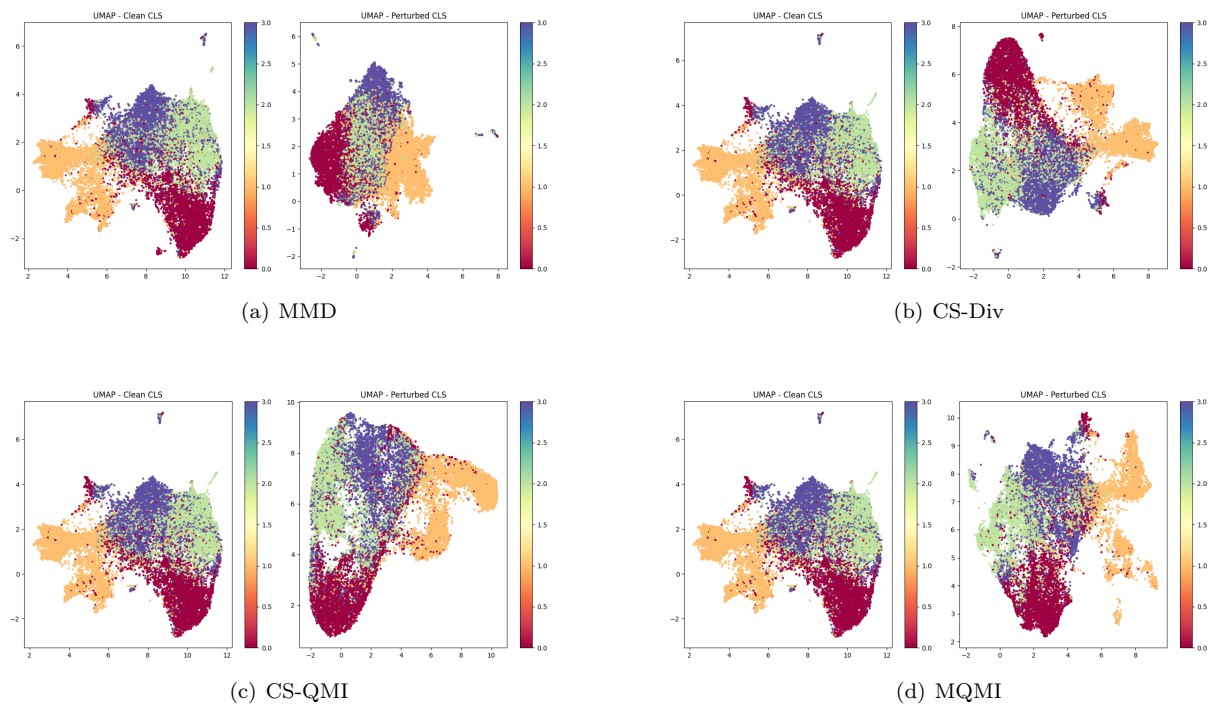

(a) MMD

(b) CS-Div

(c) CS-QMI

(d) MQMI

Figure 9: Visualization of embedding distributions using UMAP. Each panel shows **Left:** clean embeddings; **Middle:** secured embeddings; **Right:** disruption. (a) $\beta = 0.0$, where MQMI includes only the disruption term. (b) $\beta = -0.5$. (c) $\beta = -1.0$, where MQMI encourages greater diversity of disruption. The secured embeddings become more dispersed as $\beta$ decreases.

