# OpenReview forum: "Mitigating Embedding Leakage via Latent Disruption  with Controlled Reconstruction"
_TMLR — Accepted by TMLR_

### Review · Reviewer_ehWx · 2025-12-11

**Summary Of Contributions:**

This paper proposes Secured Embedding via Adversarial Learning (SEAL), which performs perturbations on embeddings in latent space to achieve protection against malicious exploitation while preserving reconstruction capabilities for trusted users. Additionally, SEAL utilizes Matrix Norm-based Quadratic Mutual Information (MQMI) to quantify dependency relationships, enhancing the method's interpretability. Experimental results demonstrate the effectiveness of SEAL.

**Audience:**

Yes

**Audience Explanation:**

In the era of large models, data privacy and security are crucial, and exploring methods to protect training data is essential for AI safety.

**Claims And Evidence:**

Yes

**Claims Explanation:**

The author provides a thorough explanation of the methodological rationale based on MQMI theory, offering sufficient elaboration on key factors such as entropy and disruption. Furthermore, the geometric motivation behind the Perturbation Term is analyzed, and extensive experimental evaluations demonstrate the method's effectiveness with sufficient persuasiveness.

**Requested Changes:**

To my knowledge, CEP [1] indicates that embedded random perturbations are beneficial for generative models. The paper analyzes and discusses this based on the paradigm of learning from noisy samples. To further enhance the strengths of this paper, I recommend that the authors conduct a discussion and comparison.

[1]  Slight Corruption in Pre-training Data Makes Better Diffusion Models. NeurIPS-2024

---

> ### Author Response · Authors · 2026-04-01
> **Response to Reviewer ehWx**
>
> We thank the reviewer for the constructive comments.
>
> **Comparison with CEP [1]**
>
> We have incorporated a discussion of CEP [1] in the revised manuscript (Page 1, Section 1, paragraph 4) to better position our work.
>
> Concretely, CEP demonstrates that controlled perturbations in embedding space can improve generative performance under benign learning objectives, acting as a form of regularization. In contrast, our work designs perturbations through the encoder and the MQMI-based learning objective to empirically reduce the exploitable information in embeddings when they are exposed to untrusted parties.
>
> This comparison clarifies that, while both approaches operate in latent space, they serve fundamentally different purposes: CEP enhances generation quality, whereas our method aims to empirically limit misuse of latent representations while preserving authorized reconstruction.
>
> ----
>
> [1] Slight Corruption in Pre-training Data Makes Better Diffusion Models. NeurIPS-2024

---

### Review · Reviewer_5z5p · 2025-12-13

**Summary Of Contributions:**

This paper studies privacy and security risks when sharing learned representations instead of raw data. It proposes SEAL, a framework that transforms a pretrained embedding $Z$ into a "secured" embedding $Z'$ via a learned latent disruption module, aiming to reduce information leakage from $Z'$. The method introduces MQMI a kernel-matrix, Renyi-2-based dependence surrogate that is optimized jointly with a private decoder that can reconstruct $\hat{X}$ from $Z'$ for authorized users. Empirically, the paper reports reductions in (i) "feature leakage" (ability to learn transferable predictors from leaked embeddings) and (ii) membership/attribute inference performance, while maintaining high-quality reconstruction through the private decoder.
Key strengths:

Clear motivation: embeddings are not inherently safe, and leakage can enable inference attacks.

A differentiable and efficient dependence objective (Renyi-2 / Frobenius norm) that avoids eigen-decomposition.

Broad experimental coverage across multiple attack types (feature leakage, membership inference, attribute inference) and modalities.

Key weaknesses:

The framework's utility is largely framed as private reconstruction, not preservation of downstream task performance on $Z'$. This is misaligned with common "share embeddings for ML tasks" workflows.

Security relies heavily on the private decoder remaining private; the paper does not convincingly close the loop under realistic leakage/adaptive attacker assumptions (e.g., partial access to $(Z',X)$ pairs or model extraction).

No formal privacy guarantees (e.g., DP-style), so the "secured" claim is primarily empirical and threat-model dependent.

Overall, I don’t think these shortcomings are specific to this paper; they reflect broader challenges in the field. At the same time, I believe this work is valuable and represents a strong contribution.

**Audience:**

Yes

**Audience Explanation:**

Representation privacy and leakage-resilient ML pipelines are relevant to parts of the TMLR audience working on privacy-preserving ML, secure representation learning, and empirical security analyses of pretrained embedding sharing. The kernel-matrix Renyi-2 objective and the evaluation of "latent leakage" are interesting and may motivate follow-up work even if the current framing and threat model need strengthening.

**Broader Impact Concerns:**

Dual use access control: A private decoder creates gating capability (authorized users can reconstruct; others cannot). This can be beneficial for privacy but may also reduce external auditability or enable undesirable control over information access. The Broader Impact Statement should discuss governance, key management, and who controls the decoder.

False sense of security: Without formal guarantees, calling embeddings "secured" may encourage deployment in high-stakes contexts where adversaries or leakage channels are stronger than tested. The paper should clearly communicate limitations and recommended deployment constraints.

Data reconstruction risk: If the private decoder or reconstruction outputs leak (logs, telemetry, insider threats), reconstructed $\hat{X}$ can reintroduce the original privacy risks. The paper should discuss operational safeguards and whether reconstruction is necessary in the intended use case.

**Claims And Evidence:**

No

**Claims Explanation:**

The experiments support the narrower claim that the proposed objective can reduce specific evaluated attacks (feature leakage and certain inference attacks) under the paper's experimental protocol, but the evidence is not fully convincing for the broader "securing representations" framing.

Main gaps:

Threat model realism / adaptive attackers: The evaluation does not sufficiently address stronger or adaptive attackers (e.g., attackers with a small amount of paired supervision $(Z',X)$, or attackers who can train decoders with auxiliary data from a similar distribution). Because the defense depends on a private decoder, leakage of decoder parameters, query access, or partial supervision should be treated explicitly.

Utility definition mismatch: Many practical settings share embeddings to enable downstream learning without reconstructing $X$. Here, utility is mainly measured via private reconstruction quality, with limited evidence that $Z'$ remains useful for downstream tasks at meaningful security levels; stronger protection may intentionally degrade task utility.

Lack of formal guarantees: MQMI is an empirical dependence surrogate; without theoretical bounds or composable guarantees, "secured" remains a heuristic claim that may not hold outside the tested settings.

Robustness/ablation clarity: The role and scale of key hyperparameters (e.g., $\beta$, kernel choices, normalization) and their privacy-utility trade-off require more systematic analysis.

**Requested Changes:**

Ref Weakness.

---

> ### Author Response · Authors · 2026-04-01
> **Response to Reviewer 5z5p**
>
> We thank the reviewer for the constructive comments.
>
> **1. Threat model limitation**
>
> We clarify the threat scenario and threat model in Sec.1 (paragraphs 1 and 4) and explicitly discuss its scope in Limitation (Third).
>
> (1) Threat model.
> We consider an adversary who has access only to exposed latent embeddings and performs inference or reconstruction (e.g., classification, membership inference, attribute inference, or public-decoder reconstruction). Stronger adversaries with additional resources (e.g., auxiliary data, paired samples, or decoder access) fall outside this setting.
>
> (2) Adaptive attackers.
> As clarified in Limitation (Third), our formulation assumes that adversaries do not have access to paired $(X, Z')$ data, auxiliary data, or the decoder itself. More powerful or adaptive attackers (e.g., with partial supervision or decoder/query access) could potentially train stronger surrogate decoders, and are not explicitly modeled in the current work.
>
> ---
>
> **2. Utility definition mismatch**
>
> We clarify that our notion of utility differs from standard embedding-sharing pipelines. In our setting, utility is defined as preserving semantically meaningful reconstruction for authorized users under an access-controlled decoder, rather than enabling downstream learning on shared embeddings.
>
> This reflects a designed trade-off: stronger protection may degrade downstream task utility, as our objective prioritizes limiting information leakage under exposure while retaining human-interpretable information through the decoder. Extending the framework to support learning-oriented utility alongside protection remains an important direction for future work, as discussed in Limitation (second point).
>
> ---
>
> **3. Lack of formal guarantees**
>
> We clarify that MQMI is not a formal mutual-information estimator with statistical guarantees. In the revised manuscript, we explicitly position MQMI as a tractable surrogate objective for reducing dependency between representations, rather than as a quantity that provides formal security guarantees (Sec.1 and Sec.3.3).
>
> More broadly, we have revised the framing of the paper to avoid over-claiming security properties. The proposed SEAL framework is described as an empirical approach that reduces exploitable information under a specified threat scenario and model, rather than as providing “secure” representations in a formal sense (Sec.1, paragraphs 1 and 4).
>
> Accordingly, our results should be interpreted as empirical evidence of mitigation under the considered setting, rather than as security guarantees that extend to stronger or unmodeled adversaries, as discussed in the limitations (page 12).
>
> ---
>
> **4. Robustness / ablation clarity**
>
> We have clarified the role and selection of key hyperparameters in the revised manuscript. In Sec. 3.2.1 (Page 4), we explain that kernel choices follow the geometry of the representations, with vMF kernels for hyperspherical embeddings and RBF kernels for chordal perturbations $\Delta$ in Euclidean space.
>
> In Sec. 4 (Page 6), we further summarize key design choices, including kernel configurations and the trade-off parameters $\beta$ and $\gamma$. In particular, $\beta$ (perturbation strength) and $\gamma$ (reconstruction weight) explicitly control the privacy–utility trade-off, and are selected via validation-based grid search over a small range. Kernel parameters follow standard practices (grid search for vMF and median heuristic for RBF). Using cosine similarity in the vMF kernel implicitly enforces unit-norm (hyperspherical) normalization of the embeddings.
>
> In Sec. 4.3 (Page 9-10), we also provide ablation results demonstrating how these parameters affect both disruption and reconstruction, highlighting dataset-dependent and non-monotonic behaviors .
>
> ---
>
>
> **5. Broad impact concerns**
>
> **(1) Governance / control.**
> We address this concern in Limitation (fourth point, page 12), where we clarify that the decoder introduces an access control mechanism and inherently imposes control over information access. This raises governance considerations, including access management, authorization, and handling of reconstructed data, which are explicitly discussed as operational concerns beyond the learning objective.
>
> **(2) Data reconstruction risk.**
> We also clarify in Limitation (fourth point, page 12) that reconstructed outputs may reintroduce sensitive information if exposed. This highlights that reconstruction itself introduces potential risk at the system level, and therefore requires appropriate operational safeguards beyond the learning objective.
>
> **(3) False sense of security.**
> We address this concern at the beginning of the Limitations section (page 12), where we explicitly state that our method provides empirical mitigation under a specific threat model, does not offer formal security guarantees, and should not be interpreted as preventing misuse in stronger or unmodeled settings.

---

### Review · Reviewer_wP1u · 2026-03-20

**Summary Of Contributions:**

The paper under review proposes a framework for latent-space data poisoning, with the goal of making leaked latent representations unsuitable for model training or feature inference. The core idea is to minimize a loss which is a sum of

(1) Matrix Norm-based Quadratic Mutual Information (MQMI) functional between source and perturbed latent

(2) Reconstruction loss for the source image/text reconstructed from the perturbed latent

for a pair of networks G (the encoder which produces perturbed latent representations) and H (private decoder which would be able to reconstruct source data from perturbed latents). The MQMI functional relies on modified form of matrix-based mutual information (which is well-known in the literature, whereas modified form is not). It is argued that this modification enables better control over the perturbations, and some theoretical justification is given connecting this modified form to the original matrix-based mutual information. Experiments on vision and text datasets demonstrate that the proposed framework reduces feature leakage and resists inference attacks compared to the baselines, whereas the private decoder is able to approximately reconstruct original data.

The paper is generally well-written even though it has shortcomings, the experiments demonstrate the strengths of the approach compared to the baselines.

**Additional Comments:**

Page 1:
"the information contained in the original signal often remains crucial (Dey et al., 2022; Chung et al., 2021)" - these references are too generic and do not properly support the argument

"Leaked embeddings, much like original data, can be exploited to train models (Zhu et al., 2024; Liu et al., 2024; He et al., 2023)" - these references, while interesting, mostly discuss data poisoning strategies and mitigations and not the argument the authors are making about leaked embeddings

"In practice, data-driven workflows increasingly rely on continuously acquired and shared information (Meyer et al., 2018)" - the work (Meyer et al., 2018) does not include continuous sharing of information (in the sense suggested in the text)

"Furthermore, they operate primarily in the input space, leaving latent embeddings vulnerable to privacy leakage." - in such a poisoned dataset, why would latents be vulnerable? References above suggest that even latents learned with contrastive learning can be poisoned making them unusable for training

Page 2:
"Our method minimizes surrogate mutual information functional to reduce dependence between original and perturbed embeddings, offering a principled and task-independent approach to limiting potential exploitation." - it would be good to say what constraints are at play, because sampling an independent latent would obviously minimize mutual information

Page 3:
"concentration near a hypersphere ∥z∥ ≃ R with radius R" - it would be clearer if a quantifier was add for R; like "with some radius R". Furthermore, reference illustrating that this behavior emerges empirically is desirable

"This can be explained by concentration of norms in high dimensions (Vershynin, 2018)" - a reference to a textbook is not very precise, a theorem or at least section is desirable

Page 4:
"In the following, we denote a normalized positive definite (NPD) matrix A" - denote or define? Also, A clearly depends on K but the notation doesn't indicate that

Eq. 1 - it is not clear whether matrix K here is the original Gram matrix or the version with vMF kernel, because notation above also includes K^Z

Eq. 2 - is A here defined in Eq. 1 for r.v. Z?

Eq. 2 - it could be instructive to write that Frobeniusm norm equals trace of matrix squared and connect it to eigenvalues

"For \Delta, where each chord" - \Delta is not defined at this point; it is only defined on Page 5: "To enable direct control over how perturbations affect the representations, we introduce the chordal shift variable ∆ := Z ′ − Z"

Page 5:
"The empirical behavior of this surrogate is further validated in our ablation study." - in the ablation study, a comparison of this surrogate with CS-QMI of Yu et al. is given for different values of \beta. Can CS-QMI be connected to direct minimization of Eq. 5 in combination with reconstruction loss? An ablation showing direct minimization of Eq. 5 + reconstruction loss would be desirable.


Page 7:
"Table 3: Ablation Study on β for MQMI." - in the ablation study, why are different range of values of \beta used in the left and the right table?

Page 9:
"our authenticator decoder" - private decoder

Page 20:
"MQMI is applied separately to the class token and all individual patch tokens, using the matrix inner product defined as ⟨A, B⟩ = trace(A^T B) for A, B \in R ^{m x n}": What matrices are plugged in here precisely?

Page 22:
In general, D.2 would benefit from more details in the calculations.

"Let Z = {z_i} follow a von Mises–Fisher (vMF) distribution with density" - are these samples independent?

"the tangent-space components δ_i^{\perp} := δ_i − (z_i^T δ_i ) z_i are isotropic with"::
a) if z_i's are random variables, why does z_i z_i^T appear in the RHS without expectations?
b) if z_i follows the vMF distribution above, the matrix z_i z_i^T would not be diagonal, so the δ_i^{\perp} won't be isotropic (unless we work in another basis, please clarify).

"For i \neq j, the cross–terms z_i^T \delta_j and z_j^T \delta_i involve the tangent components of \delta_j and \delta_i" - I think a calculation could be beneficial here

Page 23:
In general, D.3 would benefit from more details in the calculations.

Where does extra x2 scaling inside the exponential comes from in the expressiong for f_ij, g_ij, h_ij? If we take e.g. f_ij as is, it seems to conflict with the definition of A^Z later used in the expression for \| A^Z \|_F^2

"Since (z_i, \delta_i) are i.i.d. and (f_ij , g_ij , h_ij ) depend on only two such samples, the array {f_ij g_ij h_ij }_{i,j} forms an order-2 U-statistic.": not clear/not well-written
- U-statistic is not an array, it is an average over evaluations of a given function over all tuples of given size
- the exact logical implication is not properly spelled out

"exponent \beta(\rho) denotes the effective kernel scaling associated with the perturbation kernel K^{\Delta}" - it is not clear what authors precisely mean here

"During optimization both C_\kappa and the O_p (N^{−1/2}) fluctuation contribute zero gradient" - why? The terms that are now hidden in these 'constants' still have some dependency on z', and knowing that magnitude is well-controlled doesn't inform us about gradients?

Page 24:
"it serves as a surrogate measure of statistical dependence between the clean and perturbed representations" - it could be beneficial to say how well I_2(Z;Z') serves as a surrogate measure of statistical dependence

**Audience:**

Yes

**Audience Explanation:**

Protecting data against un-intended usage (i.e., data poisoning) has been a research topic in the literature; however, most of these attempts focus on input-space poisoning. Looking into data poisoning strategies in latent space is an interesting research direction.

**Broader Impact Concerns:**

No concerns about ethical implications

**Claims And Evidence:**

Yes

**Claims Explanation:**

Claims are mostly supported by the evidence.

1. Experiments demonstrate that SEAL framework can secure latent representations against un-intended use
2. Theoretical justifications need additional clarifications (Appendix D), but are convincing. Additional explanation needed for the surrogate form of matrix-based mutual information (see requested changes, points 3-4).

**Requested Changes:**

My concerns in connection to the presented paper are as follows:
1. Generally, the setting in which latent embeddings are shared *but* at the same time need to be protected against being used to train models on is not very well illustrated by the authors and the suggested references. This leads to the following questions:
a) If SEAL embeddings are public, what makes them useful in the absense of private decoder?
b) If we do not aim for usability of these embeddings in the absense of private decoder, why not use cryptographic encryption for perfect private reconstruction?
2. Mutual information doesn't change upon isomorphism, which suggests that it is impossible to achieve very high quality reconstructions with this objective; the samples provided by the authors all have perceivable visual artifacts that would make such images useless in clinical settings (see page 10). Is it possible to make these artifacts less pronounced while maintaining privacy?
3. One of the contributions is the surrogate for matrix-based mutual information, using the term -\beta \log \| A^{\Delta} \|^2_F instead of S(Z'). Analysis is provided in Appendix D, connecting this surrogate to the mutual information, but it is not clear why this surrogate is actually needed if the matrix-based mutual information is already differentiable.
4. The connection between matrix-based mutual information and the suggested surrogate is theoretically explored in Appendix D. I think some additional clarifications are needed (as suggested in the detailed feedback). Furthermore, the authors have provided numerical similations illustrating some of the steps in this derivation, but not the final result connecting two forms of matrix-based mutual information.

---

> ### Author Response · Authors · 2026-04-01
> **Response to Reviewer wP1u (1/3)**
>
> We thank the reviewer for the constructive comments.
>
>
> **1. Representation utility under adversarial exposure and the need for non-cryptographic protection**
>
> We agree that the deployment setting was not sufficiently clarified in the original manuscript.
>
> First, in our framework, perturbed embeddings are not intended to serve as directly usable representations in the absence of the private decoder. Instead, they are designed to limit downstream learning and inference under exposure. This is supported by our experiments, where models trained on perturbed embeddings fail to generalize, indicating that exploitable information is effectively suppressed. Utility is preserved only through decoder-based reconstruction for authorized users.
>
> Second, we agree that cryptographic methods provide strong guarantees for access control. However, they fundamentally differ in scope: encryption protects data before access, but does not constrain how data is used after decryption. In contrast, our framework integrates access and usage control by coupling reconstruction with a task-specific decoder. This enables controlled recoverability, where reconstruction is restricted to a specific form (e.g., human-interpretable recovery), while other forms of exploitation remain limited.
>
> We have clarified this perspective in the last paragraph of Sec. 1 (page 2).
>
> ---
>
>
> **2. Mutual information and reconstruction quality**
>
> First, we clarify that the perturbation mapping is implemented as a learnable neural network transformation (SEAL encoder), without any constraint of invertibility or bijectivity. Therefore, it should not be interpreted as an isomorphic mapping, and the invariance of mutual information under isomorphisms does not apply in our setting.
>
> Second, we clarify that MQMI is not a strict mutual information estimator, but a surrogate objective (Sec. 3.3, page 5). In particular, the coefficient $\beta$ controls the balance between its terms, so optimizing MQMI does not correspond to a well-defined change in $\mathbf I(Z;Z’)$. Even if $\mathbf I(Z;Z’)$ is reduced, this does not imply a corresponding change in $\mathbf I(X;Z’)$ along the data flow $X \rightarrow Z \rightarrow Z’$, since dependency between intermediate representations does not directly determine the information retained about the original input.
>
> Third, the observed artifacts are likely due to the current model and training choices, rather than an inherent limitation of the objective. In our experiments, we use a relatively simple decoder and standard reconstruction loss to validate the conceptual validity of the framework. In practice, reconstruction quality can be improved with more expressive decoder architectures or alternative objectives (e.g., perceptual losses). We also highlight this point in the limitations (page 11, first point), where we discuss the need for more advanced, task-aware reconstruction mechanisms.
>
> ---
>
> **3. Justification for using a surrogate instead of directly optimizing matrix-based mutual information**
>
> We agree that matrix-based mutual information is differentiable. However, directly optimizing $\mathbf S(Z’)$ only induces a global reduction of dependency and does not provide explicit control over how perturbations are applied to the representations.
>
> In contrast, our formulation introduces the perturbation variable $\Delta = Z’ - Z$ and defines the objective directly in terms of $\Delta$, enabling explicit and controllable regulation of the perturbation magnitude and structure. In particular, the parameter $\beta$ in MQMI provides direct control over the perturbation strength, allowing us to explicitly control the resulting level of disruption. This cannot be achieved by optimizing $\mathbf S(Z’)$ alone.
>
> We therefore use the surrogate not as an alternative formulation, but as a mechanism for controllable perturbation design. This motivation is further discussed in Sec. 3.3 (page 5) and Sec. 3.5 (page 6).
>
> ---
>
> **4. Towards theoretical analysis in Appendix D.**
>
> We have revised Appendix D to improve clarity by making the assumptions explicit and clarifying the approximate decomposition underlying the surrogate. We emphasize that this analysis provides a justification under stated assumptions, rather than establishing a strict equivalence between matrix-based mutual information forms.
>
> Accordingly, the derivation does not yield an exact identity that can be directly validated numerically. The numerical simulations instead support the geometric and high-dimensional behavior underlying the approximation. Additional details are addressed in the responses to specific comments.

---

> ### Author Response · Authors · 2026-04-01
> **Response to Reviewer wP1u (2/3)**
>
> **Detailed comments:**
>
>
> **Improved support for the role of original signal semantics (Dey et al., 2022; Chung et al., 2021)**
>
> We have revised paragraph 1 of the introduction (page 1) to better ground the motivation in human-in-the-loop (HITL) scenarios, and replaced the original generic references with a more targeted citation that directly supports the role of semantically meaningful information for human interpretation in machine learning and healthcare.
>
> -------
>
> **Resolution of citation mismatch for leaked embeddings claim (Zhu et al., 2024; Liu et al., 2024; He et al., 2023)**
>
> We have removed this formulation in the revised manuscript (Sec. 1, page 1) and replaced it with references that directly address privacy leakage and inference risks in latent embeddings.
>
> ---
>
> **Resolution of overly strong claim on data sharing (Meyer et al., 2018)**
>
> We agree that the original statement was too strong and not supported by Meyer et al. (2018). We have removed this formulation in the revised manuscript (Sec. 1, paragraph 2) and revised related wording in Sec. 2.2 (page 3) to ensure consistency with the cited references.
>
> ------
>
> **Resolution of ambiguity regarding input-space poisoning and latent vulnerability**
>
> In Sec. 1, paragraph 3, we clarify this distinction by separating (i) the objective of input-level poisoning methods, which focus on mitigating risks at the input level (e.g., hindering unauthorized model training), and (ii) the risks associated with leaked latent embeddings under our threat scenario.
>
> The revised formulation frames these approaches as addressing different threat scenarios and levels of risk, thereby resolving the ambiguity identified by the reviewer.
>
> -------
>
> **Clarification of constraints in the objective (Page 2)**
>
> We clarify (Sec. 1, paragraph 5) that minimizing mutual information is jointly constrained by a reconstruction objective, which prevents trivial solutions such as sampling independent latents and enforces a trade-off between disruption and reconstruction.
>
> ---
>
> **Quantification of hypersphere radius and empirical evidence (Page 3)**
>
> We revised the description in Sec. 3 to state that embeddings concentrate near a hypersphere $||z||_2 \simeq R$ for some radius $R$. We also clarify that this phenomenon is empirically observed across pretrained models and datasets, with reference to Figure 8 in the appendix.
>
> ---
>
> **More precise reference for concentration results (Page 3)**
>
> We refined the citation to Vershynin (2018) by pointing to the relevant section (Chapter 3, particularly norm concentration), improving the precision and the reability.
>
> ---
>
> **Clarification of notation and definitions for kernel-based entropy formulation (Page 4)**
>
> First, we replace “denote” with “define” and make the dependency explicit by introducing $A^Z = K^Z / \mathrm{Tr}(K^Z)$.
>
> Second, we unify the notation by consistently using $K^Z$ for the vMF Gram matrix, removing ambiguity.
>
> Third, we clarify the relation between Eq. 1 and Eq. 2 by explicitly indicating that $A^Z$ is defined from $K^Z$.
>
> Fourth, we add that $|A^Z|_F^2 = \mathrm{Tr}((A^Z)^2) = \sum_i \lambda_i^2$ to connect with its spectral interpretation.
>
> Finally, we introduce $\Delta := Z’ - Z$ at its first occurrence to ensure all variables are defined before use.
>
> ---
>
> **Clarification on comparison with direct MQMI optimization (Page 5)**
>
> We clarify that our ablation already corresponds to jointly optimizing a dependence objective (e.g., MQMI, CS-QMI, CS-Div, MMD) with reconstruction (Sec. 3.4, page 6). In particular, CS-QMI is a kernel-based quadratic mutual information estimator (closely related to Hilbert-Schmidt Independence Criterion (HSIC)), and is directly optimized together with the reconstruction loss in our experiments.
>
> While MQMI adopts a matrix-based entropy functional formulation and CS-QMI is a kernel-based dependence estimator (related to HSIC), both can be viewed as measures of dependence connected to quadratic mutual information. The comparison therefore reflects direct dependence minimization under a shared reconstruction objective.
>
> ---
>
> **Explanation of different $\beta$ ranges (Table 3)**
>
> We clarify that $\beta$ controls the perturbation–reconstruction trade-off, and its effective scale is dataset- and model-dependent. The different ranges reflect the respective operating regimes selected via validation, rather than a fixed universal interval.
>
> ---
>
> **Terminology correction for decoder (Page 9)**
>
> We have standardized the terminology to “private decoder” in the revised manuscript (Page 9) to reflect its role in access-controlled reconstruction and ensure consistency with the threat model.
>
> ---
>
> **Clarification of matrix dimensions in MQMI (Page 20)**
>
> We clarify that $A, B \in \mathbb{R}^{m \times n}$ denote token-level embedding matrices constructed from $Z$ and $Z’$, where $m$ is the number of tokens and $n$ is the embedding dimension.

---

> ### Author Response · Authors · 2026-04-01
> **Response to Reviewer wP1u (3/3)**
>
> **Response to Page 22 (Appendix D.1–D.2)**
>
> Thank you for pointing this out. We have revised Appendix D to make the assumptions and intermediate steps explicit, with the probabilistic setting clarified in D.1 and the geometric argument expanded in D.2.
>
> ---
>
> **(1) “Let $Z=\{z_i\}$ follow a vMF distribution”: are these samples independent?**
>
> We now explicitly state in Appendix D.1 that $z_1,\dots,z_N$ are i.i.d. samples from a vMF distribution on $\mathbb S^{d-1}$, and that the perturbations $\delta_1,\dots,\delta_N$ are modeled conditionally on the realized samples and are independent across $i$.
>
> ---
>
> **(2) Isotropy of $\delta_i^\perp$**
>
> We clarify in Appendix D.2 that isotropy is defined conditionally in the tangent space at $z_i$. The term $I_d - z_i z_i^\top$ is explicitly introduced as the orthogonal projector onto the tangent space $T_{z_i}\mathbb S^{d-1}$, and the covariance statement is now stated in this conditional tangent-space sense.
>
> ---
>
> **(3) Cross-terms $z_i^\top \delta_j$ for $i \neq j$**
>
> We have expanded this part in Appendix D.2 by explicitly decomposing $z_i^\top \delta_j$ into tangent and radial components. We show that the tangent contribution has conditional mean zero and variance $O(d^{-1})$, while the radial term is $O_p(d^{-1/2})$ due to near-orthogonality of independent high-dimensional samples. This makes clear why cross-terms are lower-order compared to the self-term $z_i^\top \delta_i$.
>
> Overall, Appendix D.1 now states the assumptions explicitly, and Appendix D.2 clarifies both the tangent-space formulation and the scaling behavior of the cross-terms.
>
> ---
>
>
> **Response to Page 23 (Appendix D.3–D.5)**
>
> **(1) Clarity and correctness of the derivation.**
>
> We have revised Appendix D.3–D.5 to improve clarity, correct imprecise statements, and ensure consistency across the derivation.
>
> **(a) Extra $\times 2$ scaling.**
> We clarify that the factor of $2$ arises from squaring kernel entries in the Frobenius norm expansion, i.e., $(K^{Z’}_{ij})^2$. This is now explicitly stated in Eq. (6) in Appendix D.4.
>
> **(b) U-statistic wording.**
> We replace the previous incorrect wording with the appropriate V-statistic formulation. The empirical average is now explicitly written as a second-order V-statistic (Appendix D.4, pp. 26–27), leading to the decomposition in p. 28 with explicit residual and finite-sample error terms.
>
> **(c) Meaning of $\beta(\rho)$.**
> We clarify that $\beta(\rho)$ arises as the scaling coefficient that aligns the perturbation term from the vMF expansion with the standard RBF parameterization. This is now derived and explained in Appendix D.5.
>
> **(d) Consistency with $A^Z$.**
> We revise the normalization steps (Appendix D.4, pp. 27–28) to explicitly connect the squared kernel factors with the normalized matrices $A^Z$, $A^\Delta$, and $A^{Z’}$, ensuring notational consistency throughout.
>
> ---
>
> **(2) Justification of approximations and gradient arguments.**
>
> We have removed the earlier statement suggesting that these terms contribute zero gradient. In the revised Appendix D.4–D.5, the terms $C_\kappa$ and $O_p(N^{-1/2})$ are explicitly treated as residual approximation error and finite-sample error, respectively. Rather than being ignored, they are retained as part of the approximation and reflect higher-order and sample-dependent effects in the objective.
>
> Accordingly, the optimization is carried out with respect to the leading-order terms, while $C_\kappa$ and $O_p(N^{-1/2})$ are explicitly acknowledged as residual terms arising from the approximation, rather than being ignored, but are not directly controlled in the optimization. This clarifies that the objective is derived under an approximation regime, without assuming that these components are constant or gradient-free.
>
> ---
>
> **Justification of $\mathbf{I}_2(Z;Z')$ as a dependence measure (Page 30 in Appendix E)**
>
> We clarify that $\mathbf{I}_2(Z;Z’)$ serves as a tractable measure for statistical dependence, derived from the Rényi family and implemented via matrix-based entropy functionals. It provides a differentiable and computationally efficient objective that correlates with dependency reduction in practice, as reflected in our empirical results.

---

### Review · Reviewer_Ggn8 · 2026-03-23

**Summary Of Contributions:**

This paper studies how to protect shared latent embeddings from misuse while still allowing authorized recovery of the original input. The proposed method, SEAL, learns a perturbation mechanism in latent space together with a private decoder. The core idea is to reduce dependence between the original embedding Z and secured embedding Z′ using a new Matrix Norm-based Quadratic Mutual Information (MQMI) objective, while retaining enough information for the co-trained private decoder to reconstruct the input. The method is evaluated on both vision and text benchmarks, with experiments targeting feature leakage, membership inference, attribute inference, and reconstruction quality.

Strength:

The problem is timely and important: pretrained embeddings are increasingly shared and can leak sensitive information.

The paper proposes a reasonably coherent framework that combines protection and authorized utility, rather than optimizing only one side of the trade-off.

The empirical evaluation is broad in modality coverage, spanning vision and NLP, and includes several attack-oriented evaluations and ablations.

The private-vs-public decoder comparison is a useful component of the paper’s story, since it operationalizes the notion of “authorized recovery.”

Weakness:
The theoretical justification for MQMI as a surrogate for mutual information reduction remains somewhat heuristic; the replacement of S2(Z′) by S2(Δ) is motivated geometrically, but not fully established as a faithful objective for the stated security goal.

Several evaluations are restricted to relatively simple attacker models, especially linear classifiers for privacy attacks and standard public decoders for reconstruction, so the claimed robustness against malicious exploitation may be somewhat overstated.

Some results are mixed across datasets, especially in privacy tables where gains are not uniformly strong, and the practical meaning of some metrics could be clarified further.

**Additional Comments:**

N/A

**Audience:**

Yes

**Audience Explanation:**

This work sits at an interesting intersection of representation learning, privacy, and trustworthy ML. Researchers working on embedding privacy, inversion/inference attacks, information-theoretic learning objectives, and secure deployment of pretrained encoders would likely find the paper relevant. The paper’s framing is also broader than a single modality, which increases its appeal to the TMLR audience.

I also think the “secure embeddings with authorized reconstruction” setup is a useful formulation. Much prior work either tries to preserve utility while anonymizing representations or focuses on poisoning/unlearnability in the input space. This paper instead targets latent-space sharing directly and couples protection with a private recovery pathway, which is a practically interesting angle even if the current instantiation still has limitations.

**Broader Impact Concerns:**

The work is broadly privacy-protective, so I do not see a major negative broader-impact issue that would by itself block publication.

**Claims And Evidence:**

Yes

**Claims Explanation:**

The paper provides a meaningful amount of empirical evidence for its central claims, and overall the experiments support the main conclusion that SEAL can substantially disrupt downstream exploitation of embeddings while preserving recovery for a co-trained private decoder. In particular, Table 1 shows large reductions in classification and retrieval performance from secured embeddings relative to baselines on several datasets, and Table 2 indicates weaker membership/attribute inference in a number of settings. The ablations on β and γ also help support the claimed disruption-utility trade-off, and the private-vs-public decoder comparison is aligned with the intended use case.

That said, I would not say the evidence is fully conclusive for the strongest version of the paper’s claims. The attacker models appear limited: privacy attacks are implemented with linear classifiers, and the “public decoder” comparison does not establish resistance to stronger adaptive reconstruction attacks. Likewise, the theoretical case for MQMI is suggestive but not fully rigorous; the paper explicitly presents the objective as a practical surrogate, supported by geometric motivation and ablation rather than a stronger formal guarantee. Therefore, the evidence is sufficient for the paper’s main empirical claim, but the presentation should be more careful about the scope of what has actually been validated.

**Requested Changes:**

1. Temper the security claims and define the threat model more precisely. The paper often uses broad language such as “prevents misuse” or “defends against inference attacks,” but the evidence mainly covers specific attack families and relatively simple attacker models. The claims should be narrowed to the evaluated setting, or the experiments should be expanded to include stronger adaptive attackers.

2. Strengthen the evaluation against stronger reconstruction and inference baselines. In particular, the “public decoder” comparison would be more convincing if the public adversary were given stronger architectures, more tuning budget, and possibly access to auxiliary clean data or contrastive objectives. Similarly, privacy evaluation should go beyond linear classifiers where feasible.

3. Clarify the theoretical status of MQMI. The paper should be more explicit that MQMI is a surrogate objective rather than a formal mutual-information estimator with guarantees for the security property of interest. A clearer derivation, tighter intuition, or a discussion of when replacing S2(Z′) with S2(Δ) is expected to be reliable would improve the paper substantially.

4. Improve discussion of mixed results and dataset dependence. The paper should better highlight cases where gains are modest or inconsistent, especially in some privacy metrics on text data and some feature-leakage settings.

5. Provide more implementation and hyperparameter detail in the main paper or summarize it more clearly. Since the method depends on kernels, token choices, and the β,γ trade-off, reproducibility would benefit from a clearer presentation of how these were selected.

6. Better motivate the practical deployment model. The paper should discuss what prevents an attacker from training a more capable “public” inverse model, what key material the private decoder implicitly represents, and how SEAL would be managed in a real system where data distributions evolve over time. The authors themselves note limitations around selective attribute preservation and dynamic/distributed settings; this discussion should be surfaced more prominently.

---

> ### Author Response · Authors · 2026-04-01
> **Response to Reviewer Ggn8 (1/2)**
>
> We thank the reviewer for the constructive comments.
>
> **1. Temper the security claims and define the threat model more precisely.**
>
> We agree that the original wording could be interpreted as overly broad, and we have revised the manuscript to both temper the security-related claims and more precisely define the threat model.
>
> First, we have systematically softened the language throughout the paper to avoid overgeneral claims such as “prevents misuse” or “defends against inference attacks.” These have been replaced with more precise formulations such as “empirically reduces the risk of misuse,” “mitigates embedding leakage,” and “reduces the effectiveness of evaluated attacks.” This revision has been applied consistently in the revised manuscript.
>
> Second, we have clarified both the threat scenario and the corresponding threat model in the revised manuscript (Page 1, Section 1, paragraphs 1 and 4).
>
> At the scenario level, we describe a practical setting in which latent embeddings may be exposed to untrusted parties, while mechanisms enabling human interpretation (e.g., a decoder) remain accessible only to authorized users under an access-controlled setting. This reflects real-world human-in-the-loop usage, where preserving semantically meaningful information is necessary, but also introduces potential exposure risks.
>
> At the threat model level, we define the adversary as one who has access only to the exposed latent embeddings and attempts to extract exploitable information through downstream tasks, including classification, membership inference, attribute inference, and reconstruction via a public decoder.
>
> We also note that stronger or adaptive adversaries are explicitly discussed in the limitations (Page 12, third point), where we clarify that such settings fall outside the current threat model.
>
> ---
>
>
> **2. Strengthening of the evaluation through the consideration of more powerful adversaries.**
>
> We agree that evaluating stronger adversaries is important. For inference attacks, we extend our evaluation beyond linear classifiers by including a nonlinear MLP attacker, which provides a strictly more expressive baseline and captures a broader class of decision boundaries. The full results are reported in Table 1 (Page 7).
>
> | Dataset        | Metric | Baseline | MMD   | CS-Div | CS-QMI | MQMI  |
> |----------------|--------|----------|-------|--------|--------|-------|
> | STL-10         | MLP ↓  | 94.75    | 72.54 | 75.04  | 32.42  | 10.39 |
> | Tiny-ImageNet  | MLP ↓  | 78.56    | 76.88 | 76.64  | 77.46  | 53.82 |
> | 20 Newsgroups  | MLP ↓  | 64.12    | 28.99 | 16.88  | 34.03  | 2.65  |
> | AG News        | MLP ↓  | 91.47    | 4.94  | 5.20   | 72.64  | 11.55 |
>
> For reconstruction and more adaptive adversaries (e.g., stronger architectures, increased training budget, or access to auxiliary data), we note that this would require substantially strengthening the public decoder baseline beyond the current setting. We have clarified this in the revised manuscript and explicitly discuss it as a limitation (fifth point in page 12).
>
> ---
>
>
> **3. Clarify the theoretical status of MQMI.**
>
> First, in Sec. 3.3 (page 5), we clarify that MQMI is a surrogate objective rather than a mutual information estimator with statistical guarantees. In particular, MQMI is inspired by matrix-based mutual information, but replaces the direct use of $\mathbf{S}_2(Z')$ with a surrogate formulation based on the chordal perturbation $\Delta = Z' - Z$, which explicitly isolates and controls the effect of the perturbations.
>
> Second, we further clarify the intuition in Sec. 3.5. The key observation is that $\mathbf{S}_2(Z')$ entangles two effects: the intrinsic structure of the original representations in $Z$ and the perturbation from $Z$ to $Z'$. In contrast, the chordal difference $\Delta = Z' - Z$ isolates the perturbation itself. Applying $\mathbf{S}_2(\Delta)$ therefore provides a direct way to quantify and control the perturbation-induced change, which is precisely the quantity targeted by MQMI.
>
> Third, we clarify in Sec. 3.5 and Appendix D that this surrogate is justified under an explicit high-dimensional spherical perturbation model. Under mild assumptions on the structure of the embeddings and perturbations, replacing $\mathbf{S}_2(Z')$ with $\mathbf{S}_2(\Delta)$ can be viewed as a surrogate approximation that separates perturbation effects from the original representation structure. We emphasize that this is a model-based approximation rather than a formal guarantee, and provide a detailed analysis in Appendix D.

---

> ### Author Response · Authors · 2026-04-01
> **Response to Reviewer Ggn8 (2/2)**
>
> **4. Improve discussion of mixed results and dataset dependence.**
>
> We have revised Sec. 4 (page 8–9) to provide a more explicit and balanced discussion of mixed results and dataset-dependent behavior.
>
> In Sec. 4.1 (page 8), we clarify that while MQMI achieves strong and consistent feature disruption on vision datasets, the gains on NLP tasks are comparatively more modest and less consistent. In particular, although MQMI performs strongly on 20 Newsgroups, its advantage is less consistent on AG News, where CS-Div achieves stronger reductions across multiple metrics.
>
> In Sec. 4.2 (page 9), we further highlight that improvements on text datasets are more mixed, especially for membership inference, where MQMI often does not improve over CS-Div and in some cases performs worse.
>
> In addition, we updated the ablation study in Sec. 4.3.1 and 4.3.2 (page 9) to reflect non-monotonic and dataset-dependent trends, emphasizing that the impact of hyperparameters (e.g., $\beta$ and $\gamma$) varies across datasets and evaluation metrics.
>
> Overall, these revisions make the presentation more transparent by explicitly identifying settings where MQMI is most effective, as well as cases where the improvements are modest or less consistent.
>
> ---
>
> **5. Implementation and hyperparameter clarity.**
>
> We have improved the clarity of implementation details and hyperparameter selection in the revised manuscript to enhance reproducibility. Specifically, we now clarify the kernel design in Sec. 3.2.1 (Page 4), where the choice of kernels follows the geometry of the representations: vMF kernels are used for hyperspherical embeddings, while RBF kernels are applied to the chordal perturbations $\Delta$ defined in the Euclidean space.
>
> In Sec. 4 (Page 6), we further provide a concise summary of key design choices, covering kernel configurations, token usage, and the selection of the trade-off parameters $\beta$ and $\gamma$.
>
> In particular, the vMF concentration parameter $\kappa$ is selected via grid search, while the RBF bandwidth is determined using the median heuristic. The trade-off parameters $\beta$ and $\gamma$ are selected via validation-based grid search over a small range. We also explicitly state that backbone encoders are kept frozen, and only the SEAL encoder and decoder are trained.
>
> ---
>
> **6. Clarity of practical deployment model.**
>
> We clarify the practical deployment model along several aspects.
>
> First, regarding stronger public attackers, we clarify that our current formulation operates under a restricted threat model in which adversaries only have access to perturbed latent embeddings, but not the decoder. Stronger adversaries with additional resources (e.g., more expressive models, auxiliary data, or decoder access) fall outside this scope and are explicitly discussed in the third and fifth limitations.
>
> Second, we clarify the role of the private decoder as an access-controlled component that enables reconstruction from perturbed embeddings. It is jointly learned with the perturbation mechanism, and thus can be viewed as implicitly representing the reconstruction “key” under the SEAL encoding process.
>
> Third, regarding deployment under evolving data distributions, we note that both the perturbation mechanism and the decoder may require periodic updates to remain aligned with the embedding distribution, as discussed in the final limitation.
>
> Fourth, regarding selective attribute preservation, we clarify in the Limitations (first point) that the retained information is implicitly determined by the current objective, which does not explicitly control semantic attributes; enabling fine-grained control would require extending the loss (e.g., perceptual or region-specific objectives), which remains open for future work.

---

### Decision · Action_Editor_Wz7m · 2026-05-17

**Recommendation:** Accept as is

**Audience:**

Yes

**Audience Explanation:**

This paper studies how to protect shared latent embeddings from misuse while still allowing authorized recovery of the original input, which is an important research topic in trustworthy machine learning.

**Claims And Evidence:**

Yes

**Claims Explanation:**

The authors addressed the concerns from reviewers. For the reviewer who did not respond, I checked the authors' responses and found that the authors have lowered the tone to make all claims be supported by evidence.

---

> ### Author Response · Authors · 2026-05-25
> **Acknowledgement**
>
> We thank the Action Editor and the reviewers for their constructive feedback and thoughtful suggestions, which helped us improve the paper. We also appreciate the time and effort devoted to evaluating our work.